# Introducing graupel density prediction in Weather Research and Forecasting (WRF) double-moment 6-class (WDM6) microphysics and evaluation of the modified scheme during the ICE-POP field campaign

**Sun-Young Park[1], Kyo-Sun Sunny Lim[1], Kwonil Kim[2], Gyuwon Lee[1], and Jason A. Milbrandt[3]**

[1]BK21 Weather Extremes Education & Research Team, Department of Atmospheric Sciences, Center for Atmospheric REmote sensing, Kyungpook National University, Daegu, South Korea
[2]School of Marine and Atmospheric Sciences, Stony Brook University, Stony Brook, NY, United States
[3]Environment and Climate Change Canada, Atmospheric Numerical Weather Prediction Research, Dorval, QC, Canada

**Correspondence:** Kyo-Sun Sunny Lim (kyosunlim@knu.ac.kr)

**Abstract.** The Weather Research and Forecasting (WRF) double-moment 6-class (WDM6) scheme was modified by incorporating predicted graupel density. Explicitly predicted graupel density, in turn, modifies graupel characteristics such as the fall velocity–diameter and mass–diameter relationships of graupel. The modified WDM6 has been evaluated based on a two-dimensional (2D) idealized squall line simulation and winter snowfall events that occurred during the International Collaborative Experiment for Pyeongchang Olympics and Paralympics (ICE-POP 2018) field campaign over the Korean Peninsula. From the 2D simulation, we confirmed that the modified WDM6 can simulate varying graupel densities, ranging from low values in an anvil cloud region to high in the convective region at the mature stage of a squall line. Simulations with the modified WDM6 increased graupel amounts at the surface and decreased graupel aloft because of the faster sedimentation of graupel for two winter snowfall cases during the ICE-POP 2018 campaign, as simulated in the 2D idealized model. The altered graupel sedimentation in the modified WDM6 influenced the magnitude of the major microphysical processes of graupel and snow, subsequently reducing the surface snow amount and precipitation over the mountainous region. The reduced surface precipitation over the mountainous region mitigates the surface precipitation bias observed in the original WDM6, resulting in better statistical skill scores for the root mean square errors. Notably, the modified WDM6 reasonably captures the relationship between graupel density and its fall velocity, as retrieved from 2D video disdrometer measurements, thus emphasizing the necessity of including predicted graupel density to realistically represent the microphysical properties of graupel in models.

## 1 Introduction

Over the past few decades, the parameterization of ice microphysics traditionally represents ice-phase particles as predefined categories of solid-phase hydrometeors in bulk-type cloud microphysics (Lin et al., 1983; Rutledge and Hobbs, 1983; Cotton et al., 1986; Ferrier, 1994; Meyers et al., 1997; Thompson et al., 2004; Hong and Lim, 2006; Seifert and Beheng, 2006; Morrison et al., 2009), bin-type cloud microphysics schemes (Reisin et al., 1996; Geresdi, 1998; Khain et al., 2004; Lebo and Seinfeld, 2011) and Lagrangian "super particle" microphysics schemes (Grabowski et al., 2019; Morrison et al., 2020; Shima et al., 2020). Solid-phase hydrometeors in cloud microphysics schemes are classified into typical particle types, such as ice crystals, aggregates, graupel and hail. Each category of hydrometeors is characterized by static parameters defining density, diameter–mass relationship and diameter–fall velocity relationship, which are expressed differently in each microphysics scheme. Several

studies have reported that the simulated convection was considerably sensitive to the manner of categorization of solid-phase hydrometeors (Morrison and Milbrandt, 2011; Bryan and Morrison, 2012; Adams-Selin et al., 2013). Morrison and Milbrandt (2011) demonstrated that different approaches in treating graupel or hail produce distinct differences in storm structure, precipitation and cold-pool strength for idealized supercells. This is because graupel leads to more anvil condensate and weaker cold pool compared to hail. Bryan and Morrison (2012) showed that the fall velocities of graupel and hail affect the simulated reflectivity and dynamics for an idealized squall line. Simulations with graupel instead of hail produce convective regions that are too wide and have lower reflectivity, primarily due to the slower fall velocity of graupel compared to hail. Adams-Selin et al. (2013) reported that the development of a bow echo is highly sensitive to the parameters defining the fall velocities of graupel and hail. The simulations with slower-falling graupel-like particle created a wider stratiform region and stronger cold pool, allowing for more melting and evaporation, which helped generate bowing segments earlier than in the faster-falling hail-like simulations.

Since the study of Wisner et al. (1972), research on microphysics schemes has focused on augmenting the parameterization of cold-rain processes by increasing the number of solid-phase categories or introducing new prognostic variables for these categories (Cotton et al., 1986; Ferrier 1994; Reisner et al., 1998; Milbrandt and Yau, 2005; Bae et al., 2019). More recently, modeling approaches have evolved toward ways of predicting solid-phase characteristics or considering various shapes of ice crystals (Morrison and Grabowski, 2008; Mansell et al., 2010; Milbrand and Morrison, 2013; Morrison and Milbrandt, 2015; Jensen et al., 2017; Tsai and Chen, 2020; Jensen et al., 2023). Morrison and Grabowski (2008) devised a new method that allows the changing mass–dimension and projected area–dimension relationships of ice particles to evolve according to the predicted rime mass fraction and particle dimension. Mansell et al. (2010) and Milbrandt and Morrison (2013, hereafter MM13) implemented a new approach of incorporating a prognostic graupel density. By advancing the study of MM13, Morrison and Milbrandt (2015) later developed the Predicted Particle Properties (P3) bulk microphysics scheme that predicts the rime mass fraction and rime density for a single generic ice-phase category. Jensen et al. (2017) introduced the Ice-Spheroids Habit Model with Aspect-ratio Evolution (ISHMAEL) bulk microphysics scheme, which predicts the evolution of the ice particle aspect ratio for two ice species, namely planar-nucleated and columnar-nucleated particles. Tsai and Chen (2020) proposed a bulk-type microphysics scheme that allows variations in the shape and density of solid-phase hydrometeors. Recently, Jensen et al. (2023) implemented a prognosed density graupel category into the Thompson–Eidhammer scheme (Thompson

and Eidhammer, 2014), following the approach of Mansell et al. (2010) and MM13.

Various studies have demonstrated the merits of considering the prognostic density of solid-phase hydrometeors when simulating convective storms (Johnson et al., 2016; Jouan and Milbrandt, 2019). Johnson et al. (2016) evaluated the reproducibility of the polarization signatures in supercell storms for several partially or fully two-moment (2M) schemes. Realistic signatures were obtained only with microphysics schemes that predicted graupel density. Predicted graupel density assigns high-density frozen drops to the graupel category, resulting in relatively high-density graupel that can later grow into hail. These differences in the treatment of rimed-ice processes allow hail to grow larger and produce a much more prominent hail signature. Jouan and Milbrandt (2019) demonstrated that variations in the simulated storm reflectivity and precipitation structure exhibit more pronounced differences when using predicted particle density instead of a fixed particle density in the 2M scheme, particularly related to different number concentrations of cloud condensation nuclei (CCN) in a mid-latitude continental squall line. Since CCN concentration affects cloud droplet number concentration and mean droplet diameter, the model's microphysical response depends on how well parameterized processes involving the ice phase account for droplet size effects. Mean droplet size impacts graupel growth directly through the collection efficiency between graupel and droplets. Additionally, predicted graupel density influences graupel growth by increasing graupel fall speeds and enhancing accretion rates. Based on their analysis, they suggested that an accurate representation of graupel in microphysics schemes is crucial for appropriately simulating the effects of changes in the concentration of cloud condensation nuclei in selected systems.

The Weather Research and Forecasting (WRF) double-moment 6-class (WDM6) scheme (Lim and Hong, 2010), a bulk-type microphysics scheme, has been widely evaluated for predicting deep convective precipitation in summer (Min et al., 2015; Song and Sohn, 2018; Kim et al., 2022) and snowfall events in winter (Liu et al., 2011; McMillen and Steenburgh, 2015; Morrison et al., 2015; Comin et al., 2018; Lim et al., 2020; Ko et al., 2022). Several studies have showed that the WDM6 scheme produces excess graupel compared to other microphysics schemes during the summer and winter seasons. Li et al. (2019) showed that the simulated precipitation exhibits significant sensitivity to changes in graupel density in the WDM6 scheme. Specifically, a lower graupel density tends to contribute more to 1-month precipitation amounts below 100 mm and less to those above 100 mm during the autumn season. Conversely, a higher graupel density shows the opposite pattern. Recognizing the sensitivity and importance of the representation of graupel to simulate precipitation, we introduced a new prognostic variable, the graupel volume mixing ratio, to predict graupel density based on the study of MM13. The impact of the

modified WDM6 scheme on the simulated convections was evaluated through a two-dimensional (2D) idealized squall line experiment and by considering snowfall events that occurred during the International Collaborative Experiment for Pyeongchang Olympics and Paralympics (ICE-POP 2018) field campaign over the Korean Peninsula. The novelty of our study lies in comparing the simulated graupel characteristics in the WDM6 scheme with the specialized observed data during ICE-POP 2018.

The remainder of this paper is organized as follows. Section 2 explains the implemented method of the new prognostic variable, namely the graupel density. The experimental setups, including the case description, model setup and observations for verification, are described in Sect. 3. The results and a summary are provided in Sects. 4 and 5, respectively.

## 2 New prediction variable (graupel density) in the WDM6 scheme

In the original WDM6 scheme, characteristics of hydrometeors are predefined using the static value of density ($\rho_X$) and constant coefficients for the mass ($M_X$)–diameter ($D$) and fall velocity ($V_X$)–$D$ relationships. Here, $X$ represents the species of hydrometeors, including cloud water, rain, cloud ice, snow and graupel. The specific values of parameters are available in Table A1 of the Appendix. In the WDM6 scheme, snow is defined as an unrimed ice phase (large-crystal aggregates) with a standard density of $100\,\mathrm{kg\,m^{-3}}$, indicating that it does not undergo riming. Conversely, graupel is characterized as heavily rimed crystal particles that have not undergone wet growth. In nature, graupel has a wide range of densities according to the degree of riming. However, the original WDM6 scheme is unable to simulate this variability in graupel density as it undergoes riming because it uses a predefined constant value for graupel density. This study introduces a prognostic variable, namely the volume mixing ratio ($B_G$). $B_G$ varies dynamically in both time and space, reflecting the formation and growth mechanisms of graupel. The conservation equation for $B_G$ is given by

$$\frac{\partial B_G}{\partial t} = -\mathbf{V} \cdot \nabla_3 B_G - \frac{1}{\rho_a}\frac{\partial}{\partial z}\left(\rho_a B_G V_{B_G}\right) + S_{B_G}. \tag{1}$$

The first, second and third terms on the right-hand side of Eq. (1) represent the 3D advection, sedimentation of $B_G$, and sources and sinks of $B_G$ ($S_{B_G}$). $\mathbf{V}$ and $V_{B_G}$ represent the 3D wind fields and the $B_G$-weighted mean terminal velocities of graupel, respectively; $\rho_a$ is the air density. $S_{B_G}$ comprises several microphysical source and sink processes for the mass mixing ratio of graupel ($q_G$) and density of specific hydrometeors ($\rho_X$), as defined in Eq. (2).

$$S_{B_G} =$$
$$\begin{cases} \frac{\mathrm{Piacr}}{\rho_R} + \frac{\mathrm{Praci}}{\rho_I} + \frac{\mathrm{Pracs}}{\rho_S} + \frac{\mathrm{Psacr}}{\rho_R} + \frac{\mathrm{Pgaci}}{\rho_I} + \\ \frac{\mathrm{Pgacw}}{\rho_R} + \frac{\mathrm{Psacw}}{\rho_R} + \frac{\mathrm{Pgacr}}{\rho_R} + \frac{\mathrm{Pgdep}}{\rho_G} + \frac{\mathrm{Pgfrz}}{\rho_R} \ (T < T_0) \\ \frac{\mathrm{Pgmlt}}{\rho_G} + \frac{\mathrm{Pgeml}}{\rho_G} + \frac{\mathrm{Pgevp}}{\rho_G} \ (T \geq T_0) \end{cases} \tag{2}$$

The meanings of the microphysical processes in Eq. (2) are summarized in Table 1, and their detailed descriptions are available in the literature (Appendix B of Park and Lim, 2023). $\rho_G$ can be predicted once $q_G$ and $B_G$ are updated using Eq. (3).

$$\rho_G = \frac{q_G}{B_G} \tag{3}$$

The $M_G$–$D$ relationship can be expressed as $M_G(D) = c_G D^{d_G}$. Here, $c_G$ and $d_G$ are set as $\frac{\pi \rho_G}{6}$ and 3.0, respectively, because the graupel is assumed to be a sphere in the original WDM6 scheme. Further, $c_G$ is treated as a constant since $\rho_G$ in the original WDM6 scheme is set as a constant ($500\,\mathrm{kg\,m^{-3}}$). In our modified WDM6, $c_G$ varies with the predicted $\rho_G$ (Eq. 3). The coefficients of the area ($A_G$)–$D$ relationship ($A_G = \gamma D^\sigma$), $\gamma$ and $\sigma$, are set to $\frac{\pi}{4}$ and 2.0, respectively, due to the sphere-shaped graupel in the WDM6 scheme. Mitchell (1996) addressed the fact that the Reynolds number ($Re$)–Best number ($\chi$) relationship produces the power-law expressions of fall velocity according to ice particle types based on the relationships of mass and projected area with the dimensions shown in Eq. (4).

$$Re = a_1 \chi^{b_1} \tag{4}$$

The $Re$–$\chi$ relationship was further refined by Khvorostyanov and Curry (2002) to derive the continuous power law of the ice particle dimension by adopting varying drag terms ($a_1$ and $b_1$) (Eqs. 5 and 6).

$$a_1 = \frac{C_2[(1 + C_1 \chi^{1/2})^{1/2} - 1]^2}{\chi} \tag{5}$$

$$b_1 = \frac{C_1 \chi^{1/2}}{2[(1 + C_1 \chi^{1/2})^{1/2} - 1](1 + C_1 \chi^{1/2})^{1/2}} \tag{6}$$

The non-dimensional surface roughness parameters, namely $C_1$, $C_2$, $\delta_0$ and $C_0$, in Eqs. (5) and (6) are assumed to be $4/(\delta_0^2 C_0^2)$, $\delta_0^2/4$, 5.83 and 0.6, respectively. The Best number, $\chi$, is expressed as a function of $\rho_G$ shown in Eq. (7).

$$\chi = \frac{4\rho_G g \rho_a D_{GM}^3}{3\eta^2} \tag{7}$$

Here, $g$ is the acceleration due to gravity, and $\eta$ represents the dynamic viscosity. $D_{GM}$ is the maximum dimension of the graupel. Equation (8) represents the $V_G$–$D$ relationship.

$$V_G = a_G D^{b_G} \tag{8}$$

**Table 1.** Meanings of the microphysical source and sink processes in Eq. (2).

| Symbol | Meaning | SI unit |
| --- | --- | --- |
| Paacw | Production rate for accretion of cloud water by snow or graupel | $\mathrm{kg\,k^{-1}\,s^{-1}}$ |
| Pgaci | Production rate for accretion of cloud ice by graupel | $\mathrm{kg\,k^{-1}\,s^{-1}}$ |
| Pgacr | Production rate for accretion of rain by graupel | $\mathrm{kg\,k^{-1}\,s^{-1}}$ |
| Pgacw | Production rate for accretion of cloud water by graupel | $\mathrm{kg\,k^{-1}\,s^{-1}}$ |
| Pgdep (Pgsub) | Production rate for deposition (sublimation) rate graupel | $\mathrm{kg\,k^{-1}\,s^{-1}}$ |
| Pgeml | Production rate induced by enhanced melting of graupel | $\mathrm{kg\,k^{-1}\,s^{-1}}$ |
| Pgevp | Production rate for evaporation of melting graupel | $\mathrm{kg\,k^{-1}\,s^{-1}}$ |
| Pgfrz | Production rate for freezing of rainwater to graupel | $\mathrm{kg\,k^{-1}\,s^{-1}}$ |
| Pgmlt | Production rate for melting of graupel to form rain | $\mathrm{kg\,k^{-1}\,s^{-1}}$ |
| Piacr | Production rate for accretion of rain by cloud ice (graupel) | $\mathrm{kg\,k^{-1}\,s^{-1}}$ |
| Praci | Production rate for accretion of cloud ice (graupel) by rain | $\mathrm{kg\,k^{-1}\,s^{-1}}$ |
| Pracs | Production rate for accretion of snow by rain | $\mathrm{kg\,k^{-1}\,s^{-1}}$ |
| Psacr | Production rate for accretion of rain by snow | $\mathrm{kg\,k^{-1}\,s^{-1}}$ |
| Psacw | Production rate for accretion of cloud water by snow | $\mathrm{kg\,k^{-1}\,s^{-1}}$ |

Here, $a_G$ and $b_G$ are derived from the study of Mitchell and Heymsfield (2005). By assuming the shape of graupel as a sphere, $a_G$ and $b_G$ can be expressed as shown in Eqs. (9) and (10):

$$a_G = a_1 v^{(1-2b_1)} \left( \frac{2 c_G g}{\rho_a \gamma} \right)^{b_1}, \tag{9}$$

$$b_G = b_1 (c_G - \sigma + 2) - 1, \tag{10}$$

where $v$ is the kinematic viscosity of air. Further, $c_G$ and $d_G$ represent the coefficients of the $M_G$–$D$ relationship, while $\gamma$ and $\sigma$ are the coefficients of the $A_G$–$D$ relationship, respectively. Note that $a_1$ and $b_1$ can be obtained from Eqs. (5) and (6).

$a_G$ and $b_G$ in the $V_G$–$D$ relationship are derived at the predicted $\rho_G$, which is in the range of 100–900 kg m$^{-3}$, at intervals of 100 kg m$^{-3}$ to facilitate the transition between aggregate and rime particles (Straka and Mansell, 2005), using the least-squares method in a log–log space over a range of $D_G$ of 0.3–20 mm (Table 2). Therefore, the modified WDM6 incorporates varying $a_G$ and $b_G$ parameters in the $V_G$–$D$ relationship and $c_G$ in the $M_G$–$D$ relationship by implementing predicted graupel density. Note that the coefficients, $a_G$ and $b_G$, are assumed to be 330 m$^{1-b}$ s$^{-1}$ and 0.8 in the original WDM6 scheme, and these values differ significantly from those in Table 2. However, we adhere to the methodology presented in Milbrandt and Morrison (2013) to preserve the originality of the method.

The several microphysics processes in the WDM6 can be affected by the newly derived $V_G$–$D$ and $M_G$–$D$ relationships. The microphysical processes of Pgmlt, Pgacw, Pgdep, Pgevp and Ngacw are affected by $a_G$ and $b_G$ in the $V_G$–$D$ relationship, and Pgmlt, Pgaci, Pgacr, Pgdep, Pgevp, Pgacw, Ngaci, Ngacr, Ngeml and Ngacw are affected by $c_G$ in the $M_G$–$D$ relationship. Since these processes act as a source and sink for both the mass mixing ratio and the number con-

**Table 2.** Fitted parameters of $a_G$ and $b_G$ in the graupel fall velocity ($V_G$)–diameter ($D$) relationship with varying graupel densities ($\rho_G$) (Eq. 9).

| $\rho_G$ (kg m$^{-3}$) | $a_G$ (m$^{1-b}$ s$^{-1}$) | $b_G$ |
| --- | --- | --- |
| 100 | 54.9153 | 0.5446 |
| 200 | 74.2262 | 0.5375 |
| 300 | 88.8313 | 0.5339 |
| 400 | 101.0411 | 0.5316 |
| 500 | 111.7359 | 0.5299 |
| 600 | 121.3625 | 0.5286 |
| 700 | 130.1841 | 0.5275 |
| 800 | 138.3714 | 0.5266 |
| 900 | 146.0422 | 0.5258 |

centration of cloud water, rain, cloud ice, snow and graupel (Fig. A1 in the Appendix), varying parameters with predicted graupel density can affect the mass mixing ratio and number concentration of liquid-phase hydrometeors and solid-phase hydrometeors. Figure 1 shows the retrieved $V_G$–$D$ relationship in the modified WDM6, with $\rho_G$ varying from 100 to 900 kg m$^{-3}$. The newly retrieved relationship can represent the wide range of $V_G$ with varying $\rho_G$ and $D$, unlike the relationship in the original WDM6. The modified scheme is an extension of the WDM6 scheme, and it is incorporated in the prognostic cloud ice number concentration (Park and Lim, 2023).

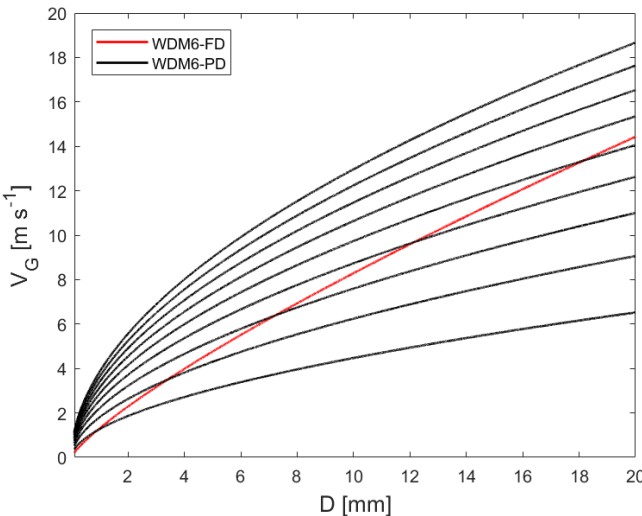

**Figure 1.** $V_G$ ($m\,s^{-1}$) as a function of $D$ (mm) with various $\rho_G$ between 100 and 900 ($kg\,m^{-3}$), utilizing $a_G$ and $b_G$ values from Table 2. The $V_G$–$D$ relationship in the original WDM6 scheme (WDM6_FD) is shown by a red line.

## 3 Experimental setup

### 3.1 Case description and model setup

#### 3.1.1 The 2D idealized squall line

The experimental design for the 2D idealized squall line simulation follows that of the study conducted by Lim and Hong (2010). A warm bubble with a 4 km radius and a maximum perturbation of 3 K at the center of the domain drives the convection. A wind of $12\,m\,s^{-1}$ is applied in the positive $x$ direction at the surface, and it decreases to 0 at a height of 2.5 km above the ground; there is no wind above this level. Additionally, no Coriolis force or friction is added, and an open boundary condition is applied for the simulation. By using the fixed initial conditions and considering only cloud microphysics parameterization as the physical option, the impact of the predicted graupel density on the simulated squall line can be distinguished and identified. The grid in the $x$ direction comprises 601 points with a grid spacing of 1 km, and 80 vertical layers are configured. The model integration duration is 6 h with a time step of 5 s.

#### 3.1.2 Snowfall during the ICE-POP field campaign

Eight snowfall events were observed during the ICE-POP field campaign period. These events can be classified into three categories (cold low, CL; warm low, WL; and air–sea interaction) according to the synoptic characteristics (Jeoung et al., 2020). Ko et al. (2022) used these eight events to compare the performances of various bulk-type microphysics schemes in simulating snowfall events. In this study, we also selected eight identical cases, following Ko et al. (2022). Ta-

ble 3 lists the model forecast and analysis periods, synoptic features, and observed accumulated precipitation (mm) for each simulation case during the analysis period. For an in-depth analysis, we selected Cases 1 and 2 as representative examples of the CL and WL categories because these two cases exhibit the most representative features of precipitation distribution for each category. Although Case 7 is listed in Table 3 as an air–sea interaction event, it is not selected for detailed analysis because only one event from this category was identified during the ICE-POP field campaign. Further details regarding the characteristics of each category are provided in the literature (Jeoung et al., 2020; Kim et al., 2021).

Figure 2 shows the accumulated precipitation amount (mm) obtained from a heated tipping rain gauge at an automatic weather station (AWS). The dot in Fig. 2 indicates the location of the MayHills Supersite (MHS; 37.6632° N, 128.6996° E; 289 m mean sea level, m.s.l.), where observation data from a 2D video disdrometer (2DVD) were collected to verify the model simulation results. These data are explained in Sect. 3.2 together with the AWS data. In the CL case, the low-pressure region is located to the north of the polar jet stream and crosses over the middle of the Korean Peninsula, leading to significant precipitation in the region (Fig. 2a). Meanwhile, in the WL case, the low pressure is positioned to the south of the polar jet stream and crosses over the southern part of the Korean Peninsula, heading toward the southeast and resulting in abundant precipitation in the coastal region (Fig. 2b).

The winter snowfall simulations during the ICE-POP 2018 field campaign were conducted using three nested domains (Fig. 3) with a horizontal grid spacing of 9, 3 and 1 km consisting of $170 \times 170$, $295 \times 349$ and $331 \times 340$ grid points, respectively. The model integration applies a one-way nesting. The top layer for the model is placed at 50 hPa, with a total of 65 vertical levels. Different integration time steps are used for each domain: 45 s for D01, 15 s for D02 and 5 s for D03. The ERA-Interim reanalysis data are used from the European Centre for Medium-Range Weather Forecasts (ECMWF) for the initial and boundary conditions (Dee et al., 2011). For physics parameterization, the Kain–Fritsch cumulus parameterization scheme (Kain, 2004) is used and applied only to the outer grid (9 km). The revised MM5 Monin–Obukhov surface layer (Jiménez et al., 2012) and the Rapid Radiative Transfer Model for General Circulation Models (RRTMG) long- and shortwave radiative schemes (Iacono et al., 2008) are used. For planetary boundary layer schemes and land surface models, the Yonsei University (YSU) (Hong et al., 2006) and Noah Multi-Parameterization (Noah-MP) models (Chen and Dudhia, 2001) are used.

### 3.2 Numerical experiments and observation data for verification

WRF version 4.1.3 (Skamarock et al., 2019) is used to simulate the 2D-idealized squall line and the wintertime snowfall

**Table 3.** Forecast and analysis periods of the selected snowfall events during the International Collaborative Experiment for Pyeongchang Olympics and Paralympics (ICE-POP) 2018 field campaign. The observed precipitation (mm) during the analysis period, obtained from the automatic weather station (AWS) by the Korea Meteorological Administration (KMA), and the synoptic features of the cases, addressed in previous studies (Jeoung et al., 2020; Ko et al., 2022), are noted.

| Case | Forecast period (UTC) | Analysis period (UTC) | Synoptic feature | Observed precipitation (mm) |
|------|----------------------|----------------------|------------------|-----------------------------|
| Case 1 | 24 Nov 2017 12:00–26 Nov 2017 12:00 | 24 Nov 2017 20:00–26 Nov 2017 00:00 | Cold low | 32.09 |
| Case 2 | 23 Dec 2017 12:00–24 Dec 2017 18:00 | 23 Dec 2017 20:00–24 Dec 2017 12:00 | Warm low | 18.6 |
| Case 3 | 22 Jan 2018 00:00–23 Jan 2018 06:00 | 22 Jan 2018 03:00–23 Jan 2018 00:00 | Cold low | 6.03 |
| Case 4 | 27 Feb 2018 18:00–1 Mar 2018 00:00 | 27 Feb 2018 23:00–28 Feb 2018 18:00 | Warm low | 57.12 |
| Case 5 | 4 Mar 2018 00:00–5 Mar 2018 12:00 | 4 Mar 2018 08:00–5 Mar 2018 09:00 | Warm low | 55.17 |
| Case 6 | 7 Mar 2018 00:00–8 Mar 2018 12:00 | 7 Mar 2018 05:00–8 Mar 2018 10:00 | Warm low | 33.07 |
| Case 7 | 15 Mar 2018 00:00–16 Mar 2018 00:00 | 15 Mar 2018 08:00–15 Mar 2018 18:00 | Air–sea interaction | 25.52 |
| Case 8 | 20 Mar 2018 12:00–21 Mar 2018 18:00 | 20 Mar 2018 18:00–21 Mar 2018 14:00 | Warm low | 25.83 |

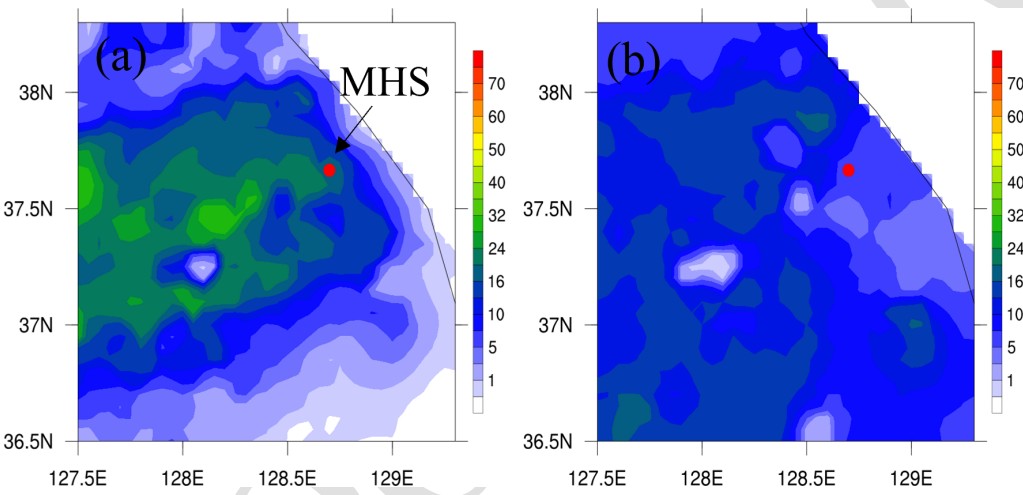

**Figure 2.** Accumulated precipitation amount (mm) during the analysis period, obtained from AWS observation for the **(a)** CL and **(b)** WL cases. The location of the observation site over the mountain, MayHills Supersite (MHS), is marked with a red dot.

cases during the ICE-POP 2018 field campaign. Two experiments, namely WDM6_FD and WDM6_PD, are conducted for each case to examine the impact of the predicted graupel density on the simulated convections. WDM6_FD uses the original WDM6 scheme with a fixed density (FD) (Lim and Hong, 2010; Park and Lim, 2023), and WDM6_PD uses the modified WDM6 scheme with predicted density (PD).

To evaluate the simulated precipitation, AWS data, from stations operated by the Korea Meteorological Administration (KMA), are used. South Korea has a total of 604 AWS surface sites. To match the horizontal resolution of the AWS, we interpolate the 1 km model simulation results into a 5 km grid. Additionally, we used the 2DVD-measured data of the diameter, fall velocity and geometry of each hydrometeor falling into a sampling area of $100\,\mathrm{cm^2}$ to validate whether the model effectively reproduces the observation-derived density–fall velocity relationship of graupel. Particle fall velocity was directly measured by the 2DVD, but particle density was estimated based on the study of Huang et

al. (2015), who adopted the Böhm method (Böhm, 1989) using the observed geometry and the 2DVD fall velocity. This method leverages the capability of the 2DVD to measure individual particles using two orthogonal cameras, making it possible to reliably estimate particle geometry, fall velocity and density. To ensure accurate measurement of the fall velocity, instances where the collocated anemometer recorded 1 min wind speeds exceeding $3.0\,\mathrm{m\,s^{-1}}$ were excluded from the analysis.

Relying solely on the 2DVD-based particle characteristics makes it challenging to differentiate graupel from other hydrometeors because of the unproven predefined assumptions on the shape, diameter and fall velocity of graupel particles in developing a hydrometeor classification algorithm. Therefore, in addition we used a collocated multi-angle snowflake camera (MASC), which captures pictures of each hydrometeor at three different angles, offering significant advantages in identifying the degree of riming and habit classification. The MASC can provide the riming index (0–1) and

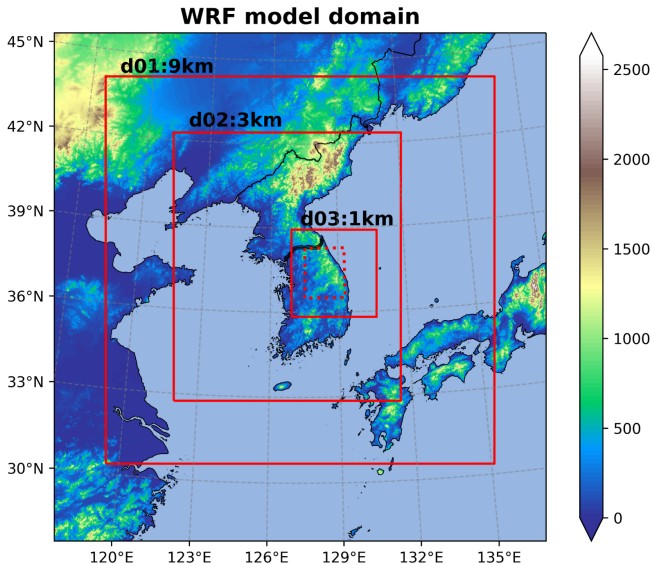

**Figure 3.** Three nested model domains with horizontal resolutions of 9, 3 and 1 km with the terrain height (m) (shaded). The dashed box denotes the analysis domain.

the complexity of the particle, which decreases as a particle becomes more spherical. These two parameters are obtained using the hydrometeor classification algorithm (Praz et al., 2017), which determines the riming index by using a pretrained supervised machine learning model and the computed geometric parameters of each particle. To identify the graupel-dominant period, the following stringent criteria are considered. The 10 min median riming index should be 1, and the 10 min median complexity of the particles should be less than 1.35. Using our criteria, we identified 11 995 graupel particles over an accumulated period of 81 min in Case 6 (Table 3).

## 4   Results

### 4.1   The 2D idealized squall line experiment

The Hovmöller plots of the maximum reflectivity and surface rainfall rate for WDM6_FD and WDM6_PD illustrate the typical evolution of a storm associated with squall line development (Fig. 4). The reflectivity is calculated using a simulated equivalent reflectivity factor, which is defined as the sixth moment of the particle size distribution based on the available mass mixing ratios and number concentrations for precipitation species including rain, snow and graupel. Both WDM6_FD and WDM6_PD simulate the strong reflectivity along the convective core region and the trailing weak reflectivity over the stratiform region, which is the general feature of squall lines (Fig. 4a and b). WDM6_PD simulates a stronger reflectivity over both convective and stratiform regions, but compared to WDM6_FD, WDM6_PD simulates

lower precipitation activities along the leading edge of the convection before 4 h (Fig. 4c and d).

The vertical distributions of the time-domain-averaged mass mixing ratio of hydrometeors for WDM6_FD and WDM6_PD and the differences between the simulations are presented in Fig. 5. The sum of the mass mixing ratios of snow and graupel is indicated by the red line. The mass mixing ratio of rain increases below the 6 km level, while that of cloud water decreases over the 4–9 km levels in WDM6_PD (Fig. 5c). Additionally, compared to WDM6_FD, WDM6_PD produces a higher snow mass mixing ratio above the 3 km level and a lower graupel mass mixing ratio over all layers. Furthermore, in WDM6_PD, the total mass mixing ratio of snow and graupel is lower below the 7 km level and higher above (Fig. 5c). Compared to the results of WDM6_FD, the generation of solid-phase hydrometeors is less effective in the lower layers and more effective in the upper layers in WDM6_PD. In contrast, the cloud ice mass mixing ratio does not show any remarkable difference between WDM6_FD and WDM6_PD.

Figure 6 shows the spatial distributions of $\rho_G$ and $q_G$, with the major source and sink microphysics processes of $q_G$ in WDM6_PD at 1 h (Fig. 6a–c), 2 h (Fig. 6d–f) and 4 h (Fig. 6g–i). Note that $\rho_G$ in WDM6_FD is predefined as $500\,\mathrm{kg\,m^{-3}}$. During the early development stage of convections, at 1 h, a graupel mass mixing ratio with relatively low density is generated over the strong updraft region, and some of the particles are transported to the upper level of 11 km (Fig. 6a and b). The main source processes contributing to the graupel mass mixing ratio are deposition (DEP), accretion (ACC) and freezing (FRZ), and the main sink processes are sublimation (SUB) and melting (MLT), as seen in Fig. 6c. Major ACC processes include the accretion process between cloud water and snow or graupel, that between rain and graupel, and that between rain and snow. At 2 h, graupel continues to be generated through DEP, ACC and FRZ, with a relatively low density of $550–800\,\mathrm{kg\,m^{-3}}$ compared to the density in the initial stage (Fig. 6a, c, d and f). The higher values of the graupel mass mixing ratios are concentrated along the updraft core, resulting in a relatively lower $\rho_G$ (Fig. 6d and e). At 4 h, graupel with a relatively lower $\rho_G$, which can be regarded as aggregation-like particles, is transported into the anvil cloud region. Over the corresponding region, DEP and ACC are the primary active processes for growing graupel.

The same microphysical properties as in Fig. 6 but for WDM6_FD are shown in Fig. 7, except $\rho_G$. Note that $\rho_G$ in WDM6_FD is predefined as $500\,\mathrm{kg\,m^{-3}}$. Throughout the simulation period, WDM6_FD produces a more abundant mass mixing ratio of graupel, reaching higher vertical levels and simulating a wider region for SUB (compare Figs. 6 and 7). At 2 h, graupel continues to be generated through DEP, ACC and FRZ, and the region with active SUB expands compared to the initial stage (Fig. 7c and d). At 4 h, more graupel is transported into the anvil cloud region at relatively

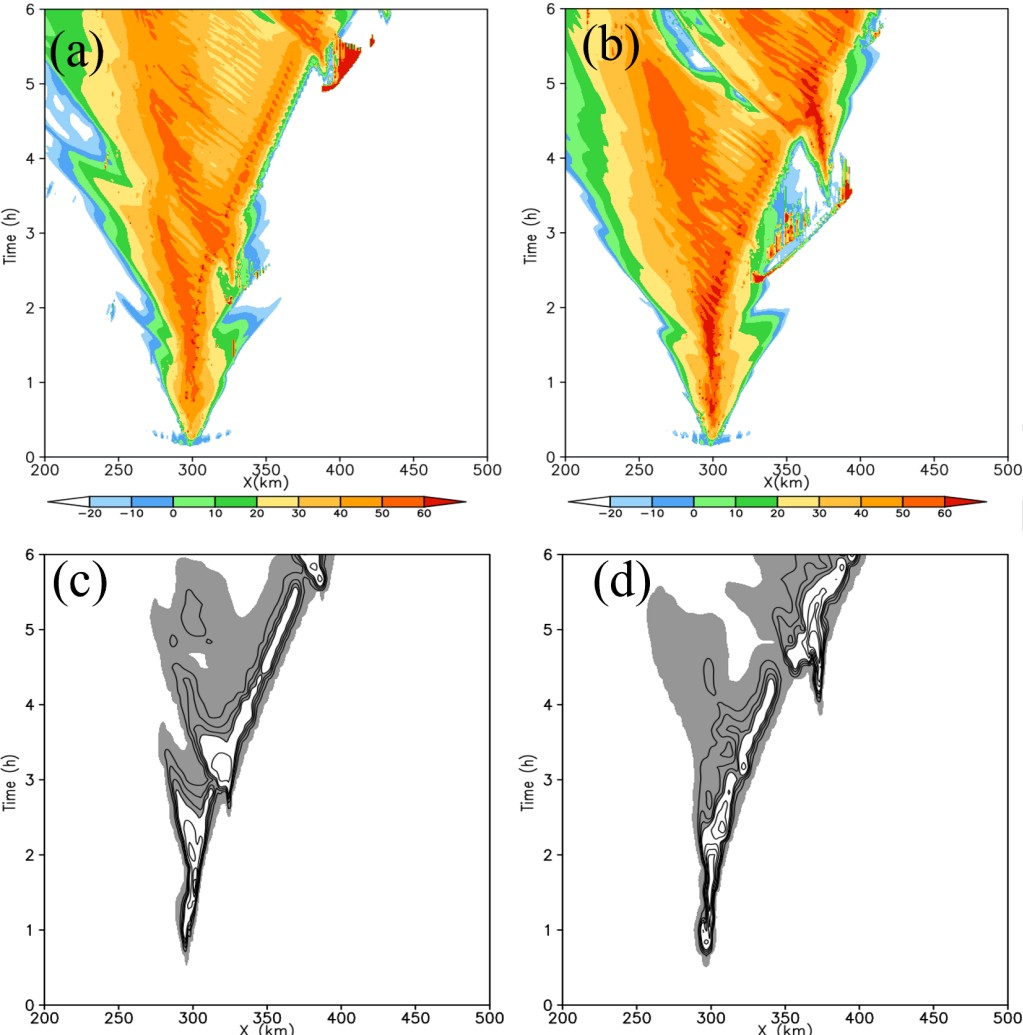

**Figure 4.** Maximum reflectivity (dBZ) for WDM6_FD and WDM6_PD is shown in **(a)** and **(b)** with the Hovmöller plots of the surface rainfall rate for **(c)** WDM6_FD and **(d)** WDM6_PD. The contour interval is 1 mm every 10 min for rates of 0–4 mm every 10 min and 3 mm every 10 min for rates greater than 4 mm every 10 min in **(c)** and **(d)**. The grey regions represent the stratiform rain region receiving precipitation at rates of 0.05–4 mm every 10 min.

lower levels compared to WDM6_FD due to active DEP and ACC in the corresponding region (Fig. 7e and f). The vertical profiles of the domain-averaged major source and sink microphysics processes are presented in Fig. S1 of the Supplement. ACC and MLT are analyzed as the most active source and sink processes in both WDM6_PD and WDM6_FD. As mentioned in Sect. 2, varying parameters with the predicted graupel density can affect the mass mixing ratio and number concentration of other hydrometeors. The spatial distribution of the mass mixing ratio of other variables (cloud water, cloud ice and snow) and the relative humidity with respect to ice (RHice) during the development stage of convection are available in Figs. S2 to S5 of the Supplement. Diao et al. (2017) suggested 125 %–130 % of the RHice threshold value is more realistic for an idealized squall line sce-

nario when compared with the National Science Foundation (NSF) Deep Convective Clouds and Chemistry (DC3) field campaign. The increase in RHice from 108 % to 130 % in our 2D squall line setup does not affect the predicted graupel density features (not shown).

## 4.2 Snowfall experiments

Figure 8 shows the simulated surface precipitation in WDM6_FD and WDM6_PD. In the CL case, most of the simulated rainfall in WDM6_PD is concentrated over the central part of the Korean Peninsula, similar to the AWS observations (Figs. 2a and 8a). The surface snow amount is similar to the surface graupel amount in both WDM6_FD and WDM6_PD in the CL case. Compared to WDM6_FD, WDM6_PD simulates less precipitation along the coast and

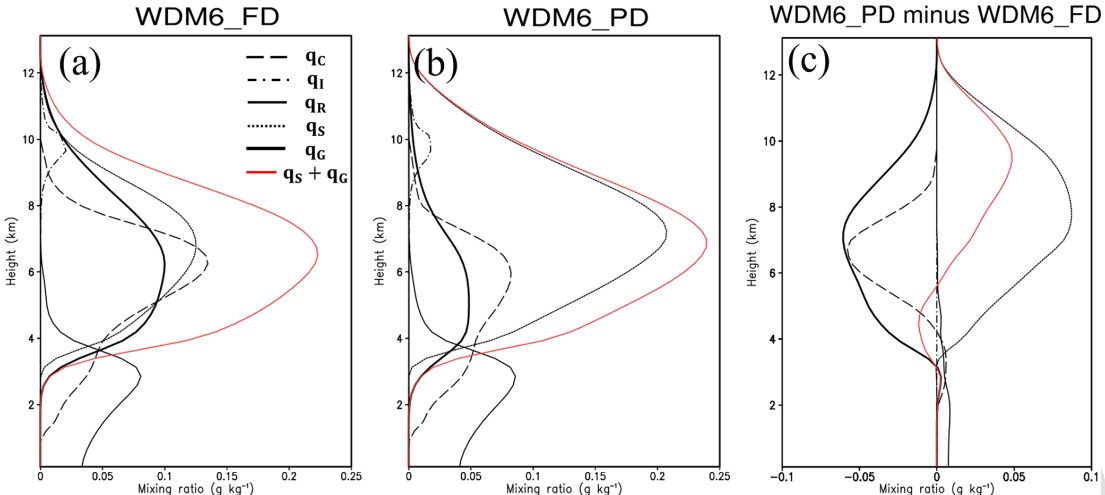

**Figure 5.** Vertical profiles of the time-domain-averaged mass mixing ratios $(g\,kg^{-1})$ of hydrometeors for **(a)** WDM6_FD and **(b)** WDM6_PD. In **(a)** and **(b)**, the cloud ice mass mixing ratio $(q_I)$ is multiplied by 10. The difference between the mass mixing ratios $(g\,kg^{-1})$ of WDM6_PD and WDM6_FD (WDM6_PD minus WDM6_FD) is plotted in **(c)**.

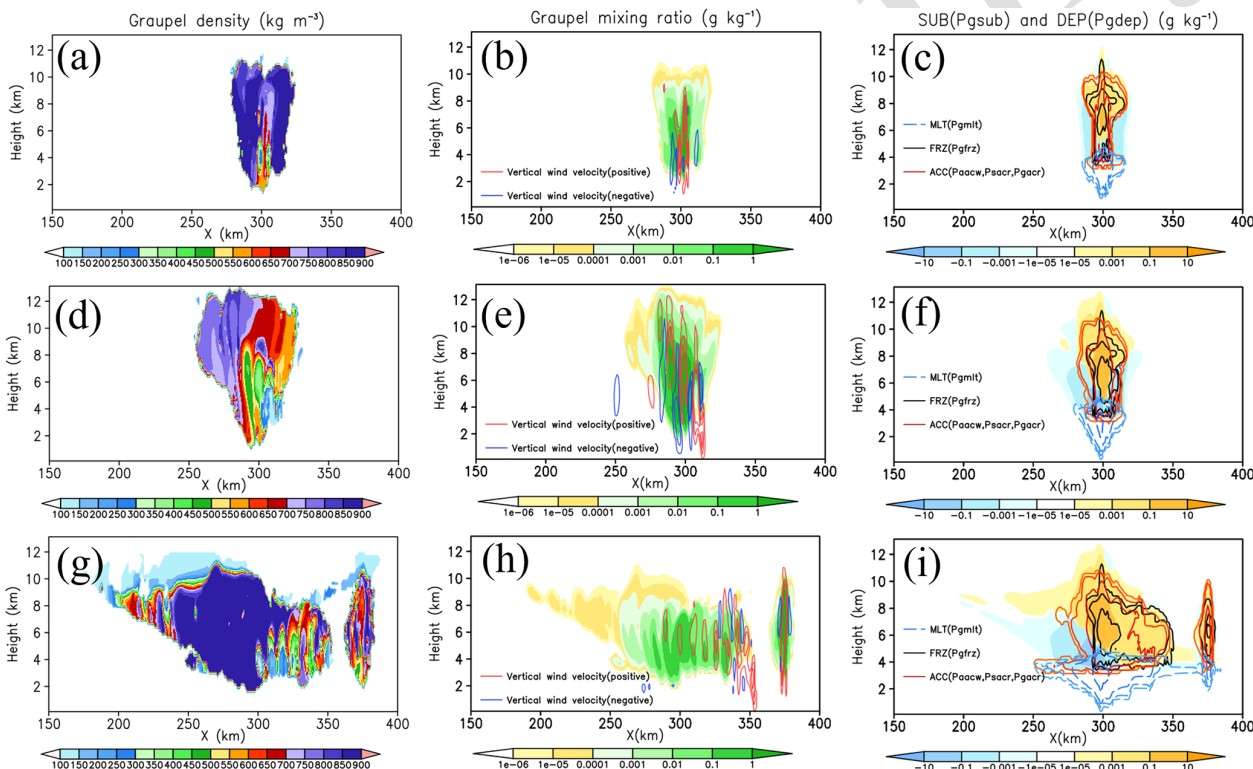

**Figure 6.** Spatial distribution of $\rho_G$ $(kg\,m^{-3})$ **(a, d, g)**, $q_G$ $(g\,kg^{-1})$ **(b, e, h)** and the major source and sink microphysics processes $(g\,kg^{-1}\,s^{-1})$ related to $q_G$ **(c, f, i)** in WDM6_PD at 1 h **(a–c)**, 2 h **(d–f)** and 4 h **(g–i)**. In **(a)**, **(d)** and **(g)**, the solid red (blue) line represents positive (negative) vertical wind velocity $(m\,s^{-1})$. Contour lines for positive (negative) values are at 2, 5 and 8 ($-2$ and $-5$) $m\,s^{-1}$. In **(c)**, **(f)** and **(i)**, the main source processes, namely deposition (Pgdep; DEP), accretion (mean of Paacw, Psacr and Pgacr; ACC) and freezing (Pgfrz; FRZ) are plotted with the major sink processes, namely sublimation (Pgsub; SUB) and melting (Pgmlt; MLT). The red (blue) colors represent DEP (SUB). The processes of FRZ, ACC and MLT are indicated by solid black, solid red and dashed blue lines, respectively. The contour lines for ACC and FRZ (MLT) values are at $1 \times 10^{-5}$, 0.001, 0.01 and 10 ($-1 \times 10^{-5}$, $-0.001$, $-0.01$ and $-10$) $g\,kg^{-1}\,s^{-1}$. Detailed descriptions of the microphysical processes are provided in Table 1.

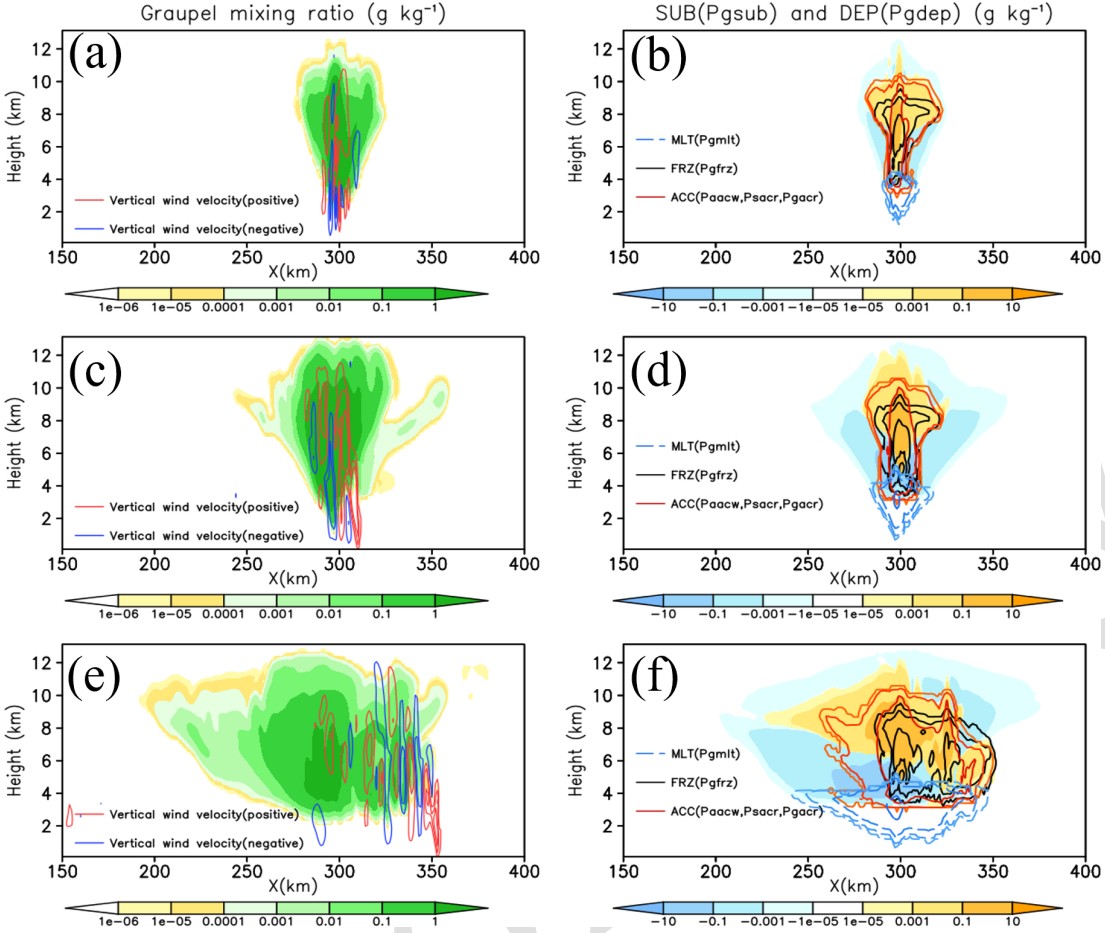

**Figure 7.** The same as Fig. 6 but for WDM6_FD.

mountainous region and more precipitation over the western part of the analysis domain (as indicated by the shading in Fig. 8b). This results in a precipitation spatial distribution that is more comparable to the observed precipitation distribution. WDM6_PD reduces the surface snow amount over the mountainous region and increases the amount of surface graupel over regions with abundant precipitation, relative to WDM6_FD (Fig. 8c and d). Specifically, the total surface snow is reduced by 93 % (domain-averaged snow amount is 0.80 mm in WDM6_FD and 0.75 mm in WDM6_PD), and surface graupel shows an increase of 124 % (domain-averaged graupel amount is 0.51 mm in WDM6_FD and 0.64 mm in WDM6_PD) in WDM6_PD compared to WDM6_FD. These changes in WDM6_PD alleviate the precipitation deficiency in WDM6_FD. Although the bias score for the CL case (Case 1) deteriorates in WDM6_PD, the root mean square error (RMSE) score for both CL cases (Cases 1 and 3) is much improved (Table 4). In the WL case, the amount of surface snow exceeds that of the surface graupel; WDM6_PD effectively alleviates the positive bias of surface precipitation, which occurs in WDM6_FD, over most of the domain (Fig. 8f). Sur-

face snow decreases significantly in WDM6_PD compared to WDM6_FD, while the surface graupel increases slightly (Fig. 8g and h). Surface snow decreases significantly by 92 % in WDM6_PD (domain-averaged snow amount is 0.84 mm in WDM6_FD and 0.77 mm in WDM6_PD) compared to WDM6_FD, while the surface graupel increases by 121 % (domain-averaged graupel amount is 0.18 mm in WDM6_FD and 0.21 mm in WDM6_PD) (Fig. 8g and h). The reduction in the surface precipitation amount in WDM6_PD results in an improvement in the RMSE scores for all WL cases, as well as biases for all WL cases except for Case 5 (Table 4). Overall, the equitable threat scores (ETSs) between the two experiments are quite similar. Despite these similar ETSs, this comparison confirms that both WDM6_FD and WDM6_PD perform comparably well in predicting snowfall events.

The vertical distributions of the time-domain-averaged mass mixing ratios for WDM6_FD and WDM6_PD are shown in Fig. 9. In the CL case, the simulated mass mixing ratios for all hydrometeors are pronounced below the 6 km level (Fig. 9a and b), while in the WL case, hydrometeors are simulated up to the 10 km level (Fig. 9d and e).

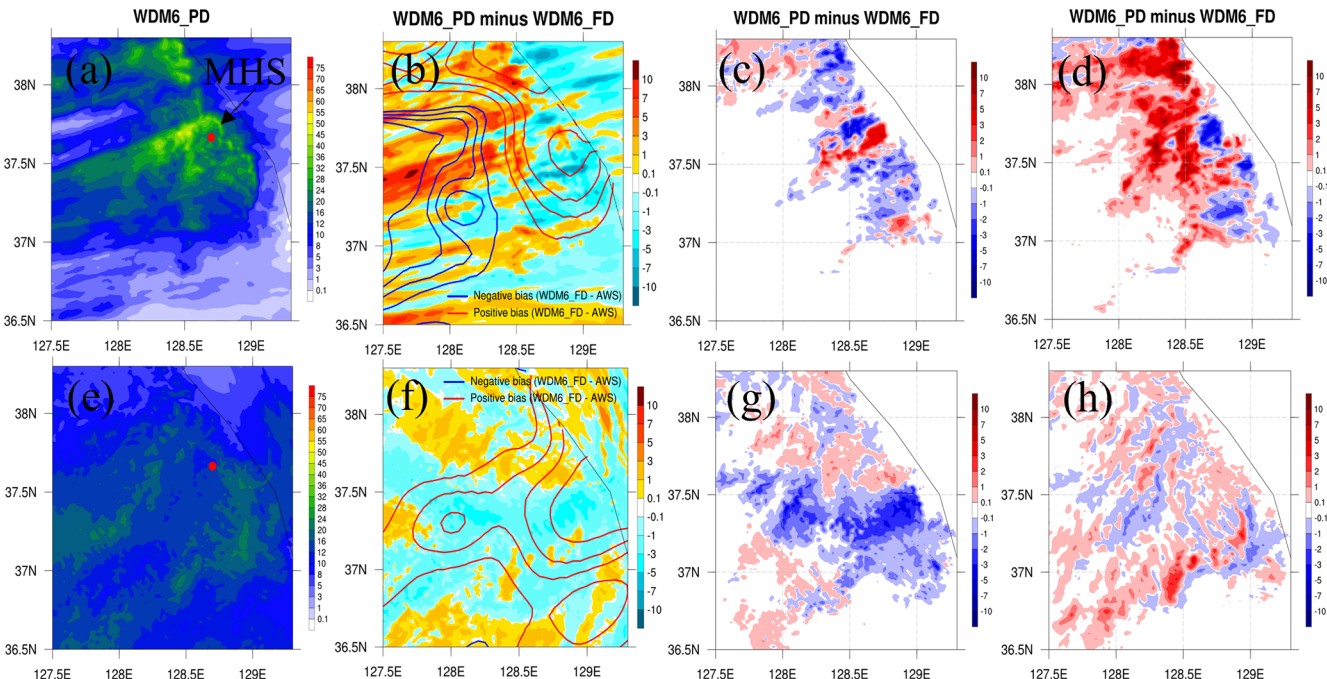

**Figure 8.** Accumulated surface precipitation amount (mm) for the **(a)** CL and **(e)** WL cases with WDM6_PD during the analysis period. The differences in the amounts of surface precipitation (mm) between WDM6_PD and WDM6_FD (WDM6_PD minus WDM6_FD) for the CL and WL cases are shaded in **(b)** and **(f)**. The red (blue) solid lines represent the positive (negative) differences between WDM6_FD and AWS observations (WDM6_FD minus AWS). The contour lines for positive (negative) values are plotted at 3, 5, 7 and 10 ($-3$, $-5$, $-7$ and $-10$) mm. The differences in the amounts of surface snow (mm) between WDM6_PD and WDM6_FD (WDM6_PD minus WDM6_FD) for CL and WL cases are plotted in **(c)** and **(g)**. The differences in the amounts of surface graupel (mm) are shown in **(d)** and **(h)**.

This is because the WL case comprises deeper systems than the CL case. The relative proportion of graupel to the total hydrometeors is greater in the CL case than in the WL case. Additionally, for the CL case, the graupel mass mixing ratio decreases, and the snow mass mixing ratio increases in WDM6_PD compared to WDM6_FD. Therefore, the total mass mixing ratio of snow and graupel increases above the 2 km level, while it decreases below the 2 km level in WDM6_PD relative to WDM6_FD in the CL case, as seen in the 2D idealized case. In WDM6_PD, the overall cloud water mass mixing ratio decreases, and the rain mass mixing ratio decreases slightly near the surface (Fig. 9c). The change in the graupel mass mixing ratio in the WL case is similar to that in the CL case (Fig. 9f). The graupel mass mixing ratio decreases significantly below the 5 km level in WDM6_PD. The snow mass mixing ratio also decreases throughout the layers, except at the 1–2 km level, resulting in a smaller total mass mixing ratio of snow and graupel in WDM6_PD compared to WDM6_FD (Fig. 9f). Moreover, the rain, cloud water and cloud ice mass mixing ratios of WDM6_FD and WDM6_PD differ only slightly. A noteworthy characteristic of WDM6_PD is the reduction in the graupel mass mixing ratio over all layers regardless of the simulation cases, resulting in an increase in the amount of surface graupel deposited (Fig. 8d and h). The reason for the lower graupel

mass (Fig. 9c and f), despite the greater surface graupel accumulation (Fig. 8d and h) in WDM6_PD, is analyzed in the subsequent Figs. 10 and 11.

The vertical profiles of the time-domain-averaged $\rho_G$ for the CL and WL cases are compared in Fig. 10. As shown in Fig. 9, convective cells develop more extensively in the vertical direction in the WL case than in the CL case. In the presence of graupel, the time-domain-averaged $\rho_G$ is simulated up to a higher level in the WL case than in the CL case (Fig. 10a and c). The value of $\rho_G$ is taken as $500\,\mathrm{kg\,m^{-3}}$ in WDM6_FD, whereas it has relatively smaller values of up to 250 and $350\,\mathrm{kg\,m^{-3}}$ in WDM6_PD for the CL and WL cases, respectively. The time-domain-averaged mass-weighted mean diameter ($D_\mathrm{m}$) in WDM6_PD is greatly reduced compared to WDM6_FD (Fig. 10b and d). In the CL case, the range of $D_\mathrm{m}$ is substantially wider below the 4 km level, indicating more variability in graupel sizes than in the WL case. In both cases, WDM6_PD presents smaller graupel than WDM6_FD, especially over the lower level. In WDM6_PD, the time-domain- and vertical-averaged $D_\mathrm{m}$ is simulated as 0.110 and 0.191 mm for the CL and WL cases, respectively, whereas in WDM6_FD, it is simulated as 0.133 mm (CL) and 0.199 mm (WL), indicating that WDM6_PD simulates smaller graupel diameters. Despite smaller values of $\rho_G$ and $D_\mathrm{m}$ in WDM6_PD compared to

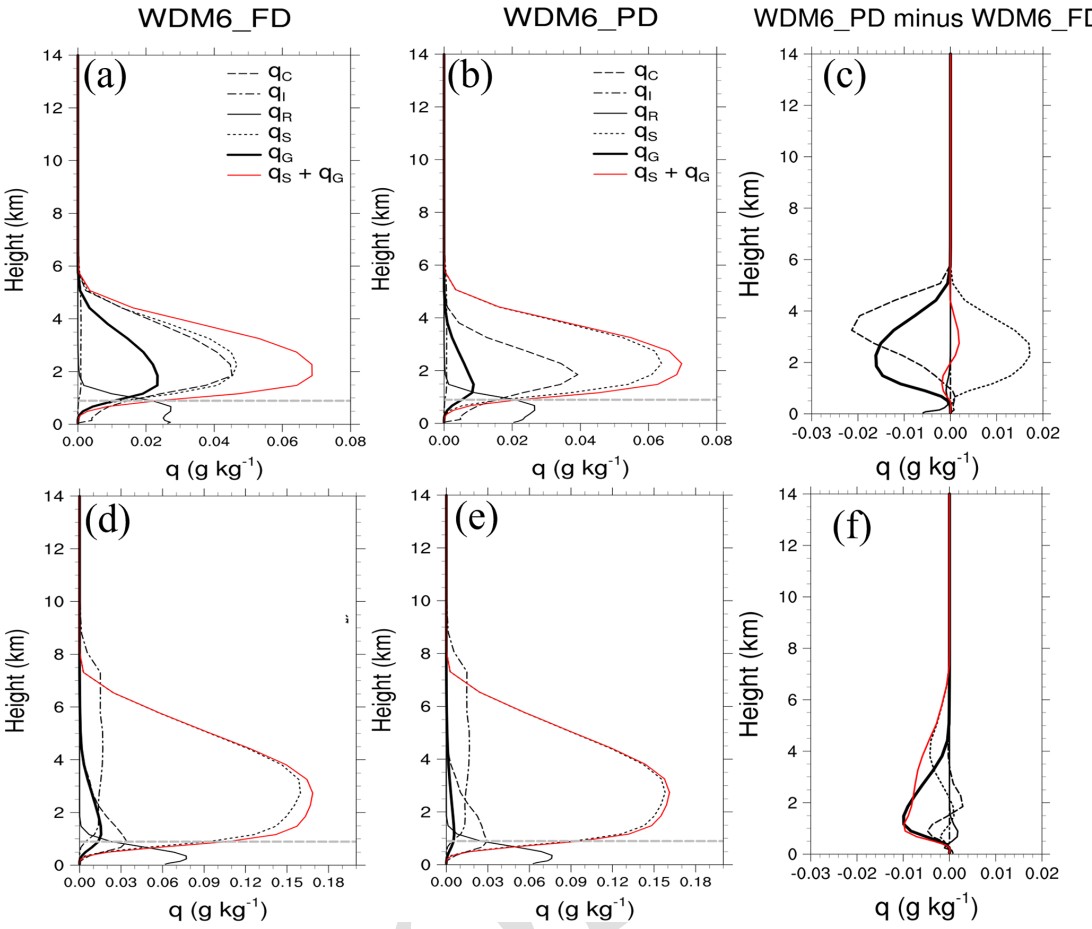

**Figure 9.** Vertical profiles of the time-domain-averaged mass mixing ratios ($g\,kg^{-1}$) of hydrometeors for the **(a)** CL and **(d)** WL cases with WDM6_FD. Panels **(b)** and **(e)** are the same as **(a)** and **(d)** but for WDM6_PD. The differences in the mass mixing ratios of WDM6_PD and WDM6_FD (WDM6_PD minus WDM6_FD) for the CL and WL cases are plotted in **(c)** and **(f)**. In **(a)**, **(b)**, **(d)** and **(e)**, the cloud ice mass mixing ratio ($q_I$) is multiplied by 100. The sum of snow and graupel mass mixing ratios ($g\,kg^{-1}$) is indicated by the red lines, and the 0° level is represented by the dashed horizontal grey line.

WDM6_FD, the former simulates a higher graupel fall velocity when considering the simulated $D_m$ in both simulations (see Fig. 1), leading to more surface graupel in WDM6_PD for the CL and WL cases (Fig. 8d and h).

In the CL case, WDM6_PD simulates $\rho_G$ with a maximum value of $220\,kg\,m^{-3}$ and $D_m$ with a maximum value of 0.44 mm at around the 2 km level (Fig. 10a and b). The maximum level of falling graupel is simulated at a lower altitude of 2 km in WDM6_PD compared to WDM6_FD, in which the maximum level is located at 3.5 km (Fig. 11a). As graupel falls quickly in WDM6_PD, graupel deposition (Pgdep) decreases, leading to the suppression of graupel growth and sublimation (Pgsub) (Fig. 11b). Moreover, the deposition of snow (Psdep) in WDM6_PD, the red lines in Fig. 11b, increases below the 3.5 km level owing to the surplus water vapor relative to WDM6_FD, leading to an increase in the snow mass mixing ratio in the atmosphere (Fig. 9c). Furthermore, the northeastern inland area, receiving abundant precipitation, exhibits more positive snow advection at the 850 hPa level in WDM6_PD compared to WDM6_FD (Fig. S6 in the Supplement). Increased snow advection toward the inland area enhances the snow mass mixing ratio in WDM6_PD. Additionally, efficient Paacw with more available snow mass can contribute to the increased snow mass mixing ratio in WDM6_PD. In the WL case, graupel, which exists up to the 10 km level, increases $\rho_G$ significantly up to a value of $350\,kg\,m^{-3}$ at the 1 km level (Fig. 10c). Even though $D_m$ of WDM6_PD is larger than that of WDM6_FD above the 3 km level, graupel particles in WDM6_PD have a greater falling velocity (Figs. 10d and 1) and fall from a relatively higher level of 8 km in WDM6_PD compared to WDM6_FD (Fig. 11c). The maximum amount of falling graupel is simulated at a relatively lower level of 1.8 km in WDM6_PD compared to WDM6_FD, as seen in the CL case. Pgdep efficiently occurs at a higher level in WDM6_PD compared to WDM6_FD (Fig. 11d), possibly because the former simu-

**Table 4.** Statistical skill scores for surface precipitation, including the root mean square error (RMSE) (mm), bias (mm) and equitable threat score (ETS) for different cases with WDM6_FD and WDM6_PD.

| Case | Experiment | RMSE (mm) | BIAS (mm) | ETS |
|------|-----------|-----------|-----------|-----|
| Case 1 | WDM6_FD | 6.58 | 1.27 | 0.30 |
|        | WDM6_PD | 6.01 | 1.61 | 0.31 |
| Case 2 | WDM6_FD | 5.49 | 5.03 | 0.16 |
|        | WDM6_PD | 4.36 | 3.56 | 0.17 |
| Case 3 | WDM6_FD | 1.81 | 1.31 | 0.19 |
|        | WDM6_PD | 1.63 | 1.26 | 0.18 |
| Case 4 | WDM6_FD | 9.51 | 2.83 | 0.07 |
|        | WDM6_PD | 9.00 | 0.63 | 0.06 |
| Case 5 | WDM6_FD | 13.95 | 12.69 | 0.14 |
|        | WDM6_PD | 13.79 | 13.27 | 0.12 |
| Case 6 | WDM6_FD | 3.94 | 2.87 | 0.10 |
|        | WDM6_PD | 3.55 | 1.31 | 0.07 |
| Case 7 | WDM6_FD | 1.67 | −1.47 | 0.10 |
|        | WDM6_PD | 1.62 | −1.36 | 0.11 |
| Case 8 | WDM6_FD | 2.63 | 1.20 | 0.17 |
|        | WDM6_PD | 1.87 | −0.36 | 0.20 |

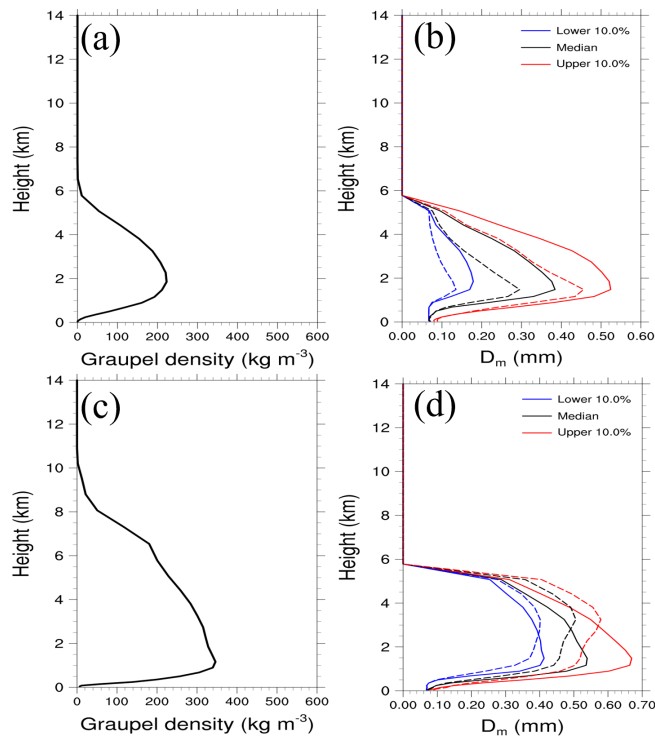

**Figure 10.** Vertical profiles of the time-domain-averaged $\rho_{\mathrm{G}}$ (kg m$^{-3}$) for the **(a)** CL and **(c)** WL cases with WDM6_PD. Time-domain-averaged $D_{\mathrm{m}}$ (mm) with WDM6_PD and WDM6_FD for the CL and WL cases is in **(b)** and **(d)**. The solid and dashed lines represent WDM6_FD and WDM6_PD, respectively.

lates more graupel with a steep increase in $\rho_{\mathrm{G}}$ between the 5 and 8 km levels. The increase in Pgdep in WDM6_PD leads to a reduction in the available water vapor, in turn causing a reduction in the Psdep and snow mass mixing ratio values in the atmosphere. The significantly enhanced graupel fall velocity, attributed to the newly derived parameters in the $V_{\mathrm{G}}$−$D$ relationship in WDM6_PD, accelerates the sedimentation of graupel. This, in turn, increases the surface graupel amount while decreasing the graupel mass mixing ratio in the atmosphere.

The $\rho_{\mathrm{G}}$−$V_{\mathrm{G}}$ relationships obtained from the 2DVD measurement at the MHS site, as well as those simulated from WDM6_PD and WDM6_FD, are shown in Fig. 12. The observed $\rho_{\mathrm{G}}$ values are in the range of 43.6–1267 kg m$^{-3}$ (Fig. 12a). The maximum normalized frequency of the observed $\rho_{\mathrm{G}}$ is shown in the range of approximately 300–400 kg m$^{-3}$, with the frequent normalized frequency of $\rho_{\mathrm{G}}$ values between 100 and 400 kg m$^{-3}$. WDM6_FD only presents a single value of $\rho_{\mathrm{G}}$ (500 kg m$^{-3}$; Fig. 12b), as it is treated as the fixed value in the model and shows a much lower range of graupel fall velocity than the observed value. In WDM6_PD, the range of $\rho_{\mathrm{G}}$ is simulated from 100 to 900 kg m$^{-3}$, as our study sets the possible range of $\rho_{\mathrm{G}}$ within this range. WDM6_PD presents the majority of simulated $\rho_{\mathrm{G}}$ at relatively lower values of 150 kg m$^{-3}$ compared to the observed value (Fig. 12a and c). The fall velocity of graupel, varying with $\rho_{\mathrm{G}}$, shows a relatively larger value in WDM6_PD than in the observations. Although WDM6_PD

simulates larger ranges of fall velocity and lower ranges of $\rho_{\mathrm{G}}$, it is closer to the observations than WDM6_FD. Our analysis highlights that WDM6_PD with varying graupel densities results in faster fall velocities, leading to more efficient sedimentation processes, which affect the spatial distribution and amount of graupel mass mixing ratio both in the atmosphere and on the surface. By predicting graupel density, WDM6_PD can produce more realistic characteristics of graupel particles, including their density and fall velocity.

## 5    Summary and conclusion

This study introduces a method to predict graupel density and incorporates the predicted graupel density into the WDM6 microphysics scheme (Park and Lim, 2023). By using the new prognostic variable (graupel volume mixing ratio), graupel density can be predicted based on the ratio of the graupel mass mixing ratio and its volume mixing ratio, following the study of Milbrandt and Morrison (2013). Therefore, the mass–diameter and fall velocity–diameter relationships of graupel are updated with varying graupel densities. To assess the impact of the predicted graupel density on the simulated precipitation system, numerical simulations are conducted for 2D idealized squall line and winter snowfall cases dur-

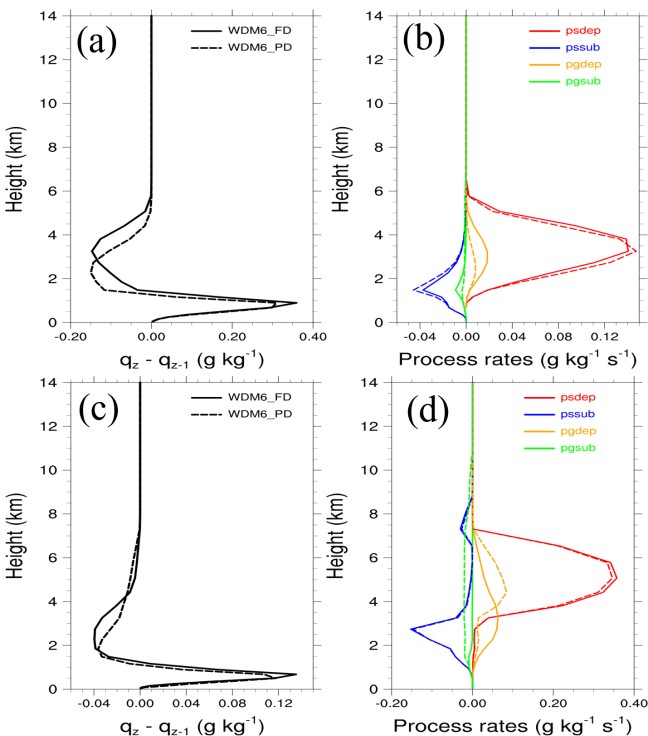

**Figure 11.** Time-domain-averaged difference in the graupel mass mixing ratio between the $z$ ($q_z$) and $z-1$ ($q_{z-1}$) levels due to sedimentation in **(a)** and **(c)** for the CL and WL cases. Panels **(b)** and **(d)** show the vertical profiles of time-domain-averaged source and sink processes of the graupel and snow mass mixing ratios for the CL and WL cases. The solid and dashed lines represent WDM6_FD and WDM6_FD_500, respectively. Only the major microphysical processes are represented. In **(d)**, Pgdep and Pgsub are multiplied by 10.

ing the ICE-POP 2018 field campaign using the WRF model version 4.1.3. The modified WDM6 requires 22.8 % TS1 more computational time, considering only cloud microphysical processes, compared to the original WDM6.

5  In the idealized 2D squall line framework, simulations using the original WDM6 and modified WDM6 yield similar surface rain rates associated with squall line development. However, compared to the original WDM6, the modified WDM6 gives higher maximum reflectivity in both the con-
10 vective cores and the stratiform regions. A comparison of the vertical profiles of the mass mixing ratios with the modified and original WDM6 confirms a significant decrease in the graupel mass mixing ratio and an increase in the snow mass mixing ratio throughout the vertical layers. The vertical cross
15 sections of graupel fields over time reveal that the modified WDM6 can represent a range of graupel densities, from low to high at varying times and in different spaces. For the graupel mass mixing ratio, the main source processes are considered to be deposition, accretion and freezing, while the
20 sink processes are considered to be sublimation and melting throughout the squall line evolution.

For the winter snowfall cases during the ICE-POP 2018 field campaign, the original WDM6 exhibits a positive bias by simulating more precipitation along the coastal and mountainous regions, irrespective of the specific case. In a shal-  25
low system, classified as a CL case in our study, the modified WDM6 provides a better RMSE score than the original WDM6 by reducing surface precipitation over the regions representing positive bias and enhancing it over the western part of the analysis domain. Although the maximum density  30
of graupel in the modified WDM6 is smaller than that in the original WDM6, the fall velocity of graupel is greater in the modified WDM6 because of the newly employed graupel fall velocity relationship. Faster sedimentation of graupel leads to inefficient graupel deposition. This, in turn, results in a de-  35
crease in the graupel mass mixing ratio and presence of more snow suspended in the atmosphere. The increased snow is a result of efficient snow deposition with surplus water vapor. Therefore, a decrease in surface snow over the mountainous region and an increase in surface graupel over regions with  40
abundant precipitation mitigate the surface precipitation deficiency in the original WDM6.

In the deep system, classified as a WL case, the modified WDM6 reduces surface snow to mitigate the excessive precipitation bias observed in the original WDM6 simulation  45
over the entire domain. In this case, the surface amounts of snow exceed those of graupel, unlike in a CL case where the simulated amounts of surface snow and graupel are similar. Therefore, the change in surface precipitation is mainly attributed to changes in the surface snow. A greater graupel  50
deposition in the 4–8 km level in the modified WDM6 consumes more water vapor, leading to inefficient snow deposition in the corresponding level. Hence, the snow mass mixing ratio in the atmosphere and at the surface decreases in the modified WDM6, leading to improved RMSE scores in  55
all WL cases compared to the original WDM6.

The simulated fall velocity–density relationship of graupel is verified using 2DVD measurement data for a WL snowfall case that occurred during the ICE-POP 2018 field campaign. Although the modified WDM6 simulates slightly  60
larger ranges of fall velocity and lower ranges of graupel density, it captures the observed relationship between graupel density and fall velocity fairly well. In contrast, the original WDM6, with a fixed graupel density, not only underestimates the graupel fall velocities but also predicts a lower  65
range of fall velocity compared to the observed values. It is worth noting that our study is distinguished by its attempt to compare simulated graupel characteristics with observed data during ICE-POP 2018. The co-located MASC measurements, coupled with the 2DVD measurement, enhance the  70
quality of graupel identification in our research. The $V_G$–$D$ relationship in the modified WDM6 is derived using the least-squares method in a log–log space at the given graupel density. The derived $V_G$–$D$ relationship in our research could be refined by incorporating a broader range of graupel obser-  75
vational data, including hexagonal, conical, lump graupel or

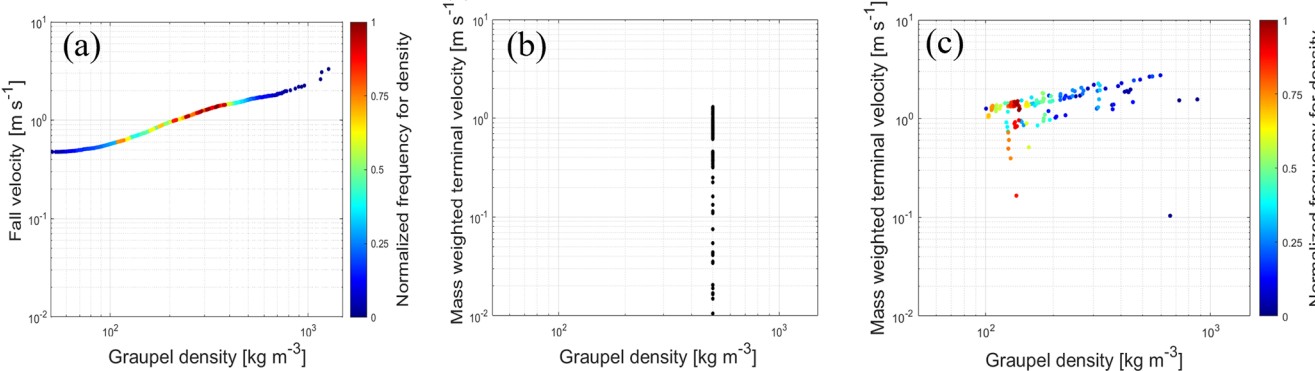

**Figure 12.** $\rho_G$–$V_G$ relationships are shown: **(a)** 2DVD measurement, **(b)** WDM6_FD and **(c)** WDM6_PD. The color bars in **(a)** and **(b)** represent the normalized frequency of $\rho_G$. In **(a)**, graupel particle characteristics measured at the MHS site during the analysis period of Case 6 are used. For **(b)** WDM6_PD and **(c)** WDM6_FD, model-simulated graupel characteristics are extracted over 16 grid points centered on the MHS site during the analysis period for Cases 2 and 6.

graupel-like snow. Improvements in the representation of the $V_G$–$D$ relationship can lead to better simulation of precipitation and microphysical processes in environments where various types of graupel are generated. Additionally, the potential benefits of the predicted graupel density could be further evaluated in future work through comparison with additional observational data such as sonde and satellite.

## Appendix A: Description of WDM6 microphysics scheme

### A1 Parameters for hydrometeor characteristics

The WDM6 microphysics scheme was originally described in Park and Lim (2023). It employs the double-moment approach for the mass mixing ratio ($q_x$) of $X = \{c, r, i, s, g\}$ and the total number concentration ($N_x$) of $X = \{c, r, i\}$. Here, c, r, i, s and g indicate cloud, rain, cloud ice, snow and graupel, respectively. The characteristics of hydrometeors in the WDM6 scheme are determined by density ($\rho_x$), the fall velocity ($V_x$)–diameter ($D$) relationship, the mass ($M_x$)–$D$ relationship and the size distribution ($N_x(D)$). The size distribution of each hydrometeor category $X$, except cloud water, which does not undergo sedimentation, is represented by a complete gamma function of the following form:

$$N_X(D)\left[\mathrm{m}^{-4}\right] = N_{0X} D^{\mu_X} \exp\{-(\lambda_X D)\}, \tag{A1a}$$

$$N_{0X}\left[\mathrm{m}^{-4}\right] = N_X \lambda_X^{\mu_X+1}, \tag{A1b}$$

$$\lambda_X[\mathrm{m}^{-1}] = \left[\frac{c_X N_X}{\rho_a q_X} \frac{\Gamma(\mu_X + d_X + 1)}{\Gamma(\mu_X + 1)}\right]^{1/d_X}. \tag{A1c}$$

Here, $N_x(D)$ indicates the number concentration of each hydrometeor corresponding to $DD_I$. $\mu_X$ and $\lambda_x$ represent the shape and slope parameters of the size distribution. $N_{0X}$ and $N_x$ are the intercept parameter and the total number concentration of each hydrometeor, respectively.

Moreover, the $V_x$–$D$ and $M_x$–$D$ relationships can be expressed as Eqs. (A2) and (A3):

$$V_x\left[\mathrm{m\,s}^{-1}\right] = a_x D^{b_x}, \tag{A2}$$

$$M_x\left[\mathrm{kg}\right] = c_x D^{d_x}, \tag{A3}$$

where $a_x$, $b_x$, $c_x$ and $d_x$ are coefficients that can vary depending on the type of hydrometeor. All particles in the original WDM6 scheme are assumed to be spherical with constant bulk densities. Thus, for each category, $c_x = \pi \rho_x / 6$ and $d_x = 3$. The coefficients defining the characteristics of hydrometeors in the original WDM6 scheme are summarized in Table A1.

### A2 Microphysical processes

The governing equations of the mass mixing ratio and the number concentration for each hydrometeor are given by Eqs. (A5) and (A6), respectively:

$$\frac{\partial q_x}{\partial t} = -\boldsymbol{V} \cdot \nabla_3 q_x - \frac{1}{\rho_a} \frac{\partial}{\partial z} \left(\rho_a q_x V_{q_x}\right) + S_{q_x}, \tag{A4}$$

$$\frac{\partial N_x}{\partial t} = -\boldsymbol{V} \cdot \nabla_3 N_x - \frac{1}{\rho_a} \frac{\partial}{\partial z} \left(\rho_a N_x V_{N_x}\right) + S_{N_x}, \tag{A5}$$

where the first and second terms on the right-hand side of Eq. (A4) represent the 3D advection and sedimentation for $q_x$, respectively. The third term represents the source and sink of $q_x$. $\boldsymbol{V}$ and $V_{q_x}$ represent the 3D wind fields and the $q_x$-weighted mean terminal velocities of $X$, respectively; $\rho_a$ is the air density. Equation (A5) is identical to Eq. (A5) but for the number concentration.

The production terms ($S_{q_x}$ and $S_{N_x}$) for each hydrometeor category are composed of several microphysical processes, including melting, accretion and nucleation, as shown

in Fig. A1. One of the accretion processes, Psacr, represents the accretion between snow and rain particles, which primarily contributes to the formation of graupel or snow. When the mass mixing ratios of both rain and snow are greater (smaller) than $1 \times 10^{-4}$ kg kg$^{-1}$, it contributes to the formation of graupel (snow). This process acts as a source process for the graupel or snow mass mixing ratio and as a sink process for the rain mass mixing ratio (Fig. A1a). Detailed descriptions and parameterization equations of these microphysical processes are available in previous studies by Park and Lim (2023) and Lim and Hong (2010).

**Table A1.** Parameters for hydrometeor (rain, ice, snow and graupel) characteristics in the WDM6 scheme.

| | $V_x-D_x$ relationship | | $M_x-D_x$ relationship | | Shape parameter $(\mu_x)$ | Density $(\rho_x)$ |
|---|---|---|---|---|---|---|
| | $a_x$ | $b_x$ | $c_x$ | $d_x$ | | |
| Rain | 841.9 | 0.8 | $\frac{\pi \rho_R}{6}$ | 3 | 1 | 1000 |
| Cloud ice | $2.71 \times 10^3$ | 1.0 | $\frac{\pi \rho_I}{6}$ | 3 | 0 | 500 |
| Snow | 11.72 | 0.41 | $\frac{\pi \rho_S}{6}$ | 3 | 0 | 100 |
| Graupel | 330.0 | 0.8 | $\frac{\pi \rho_G}{6}$ | 3 | 0 | 500 |

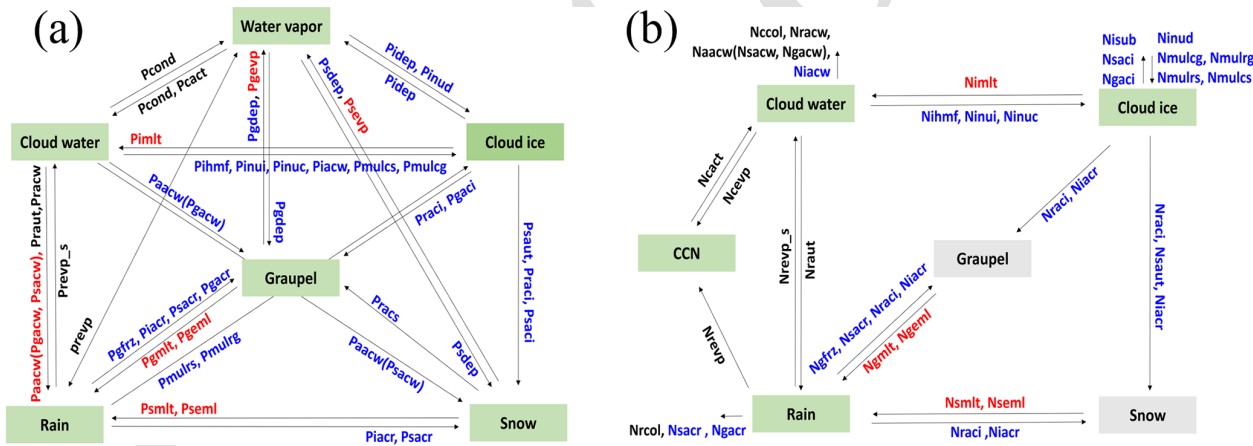

**Figure A1.** Flowcharts of microphysical processes for predicting the **(a)** mass mixing ratio ($S_{q_x}$) and **(b)** number concentration ($S_{N_x}$) of hydrometeors in the WDM6 scheme. The number concentrations of hydrometeors in the green boxes are predicted only (e.g., cloud water, cloud ice, rain and cloud condensation nuclei (CCN)). Microphysical terms in red (blue) are activated when the temperature is above (below) 0°. Terms in black are activated regardless of temperature.

*Code and data availability.* The source code of the Weather Research and Forecasting model (WRF v4.1.3) from the National Center for Atmospheric Research (NCAR) is available at https://doi.org/10.5065/D6MK6B4K [TS2] and https://github.com/wrf-model/WRF/releases (last access: January 2022). The ERA-Interim reanalysis data from the European Centre for Medium-Range Weather Forecasts (ECMWF) for initial and boundary conditions is available at https://www.ecmwf.int/en/forecasts/dataset/ecmwf-reanalysis-interim [TS3] (last access: June 2023). The model codes, model output and scripts that cover every data and figure processing action for all the results reported in this paper are available at https://doi.org/10.5281/zenodo.12065447 (Park and Lim, 2024). The 2DVD data are available at https://doi.org/10.5281/zenodo.10126522 (Kim et al., 2023).

*Supplement.* The supplement related to this article is available online at: https://doi.org/10.5194/gmd-17-1-2024-supplement.

*Author contributions.* SP designed and performed the model simulations and analysis under the supervision of KL. KL and SP wrote the paper with substantial contributions from all co-authors. KK processed the observational data. JAM provided the code to predict the prognostic volume mixing ratio of graupel. KL, GL and JAM contributed to the scientific discussions and gave constructive advice.

*Competing interests.* The contact author has declared that none of the authors has any competing interests.

ther geographical representation in this paper. While Copernicus Publications makes every effort to include appropriate place names, the final responsibility lies with the authors.

*Acknowledgements.* The authors are greatly appreciative of the participants of the World Weather Research Programme Research Development Project and Forecast Demonstration Project, as well as the International Collaborative Experiments for Pyeongchang 2018 Olympic and Paralympic (ICE-POP 2018) winter games, hosted by the Korea Meteorological Administration.

*Financial support.* This research was supported by the National Research Foundation of Korea (NRF) grant funded by the Korean government (MSIT) (RS-2023-00208394).

*Review statement.* This paper was edited by Po-Lun Ma and reviewed by Minghui Diao and two anonymous referees.

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
