# Peer review of "Introducing Graupel Density Prediction in Weather Research and Forecasting (WRF) Double-Moment 6-Class (WDM6) Microphysics and Evaluation of the Modified Scheme During the ICE-POP Field Campaign"

_Geoscientific Model Development, 2023_

## Referee Comment (RC1)

**Overview**
This study describes the introduction of prognostic graupel bulk volume (and thus predicted graupel density) into a double-moment bulk microphysics scheme (WDM6) and evaluates its impact on (1) a 2D idealized squall line simulation and (2) simulations of several observed cases during a field campaign over the Korean Peninsula. The latter simulations are compared with observations including a large spatial array of surface meteorological stations that measured precipitation, a 2D video disdrometer, and a multi-angle snowflake camera. The study found that introducing predicted graupel density improved the representation of surface precipitation spatially and reduced statistical errors in most of the case study simulations, and that the new scheme reasonably represented the observed relationship between graupel density and fall velocity. While this study is founded on sound science questions and a robust methodology, and provides some very interesting results, there are several aspects that need to be addressed and improved before being considered for publication. Overall, the manuscript and interpretation of results could be improved by a more thorough description of the microphysics scheme as opposed to just the description of the new implementation. Since the study is heavily focused on a microphysical evaluation, there is justification to provide a little more background on the scheme structure. This study obviously involved a significant amount of work, but I am skeptical of some of the physical interpretations of the results. Apart from that, there are important and interesting results that are presented, and I think the manuscript would be particularly improved by focusing more closely on those robust results instead of casting such a wide net on the evaluation. The implementation of the new scheme and the vast constraints used to evaluate it against the observations are a huge undertaking that was performed well, and I think focusing on the observation-model comparison more would better highlight the novelty and success of the science that was performed.

**General/Major Comments**
- Overall, the description of the scheme needs to be revised/revisited. While some existence of knowledge should be assumed by the reader, the paper would benefit greatly from some additional information–even just a few basic sentences on the foundation of the WDM6 scheme. In addition, an improved description of the implemented, modified graupel species is needed, in particular how this implementation affects (or doesn't affect) the other ice species in the scheme.
- In general, much of the introduction was characterized by referencing past studies saying that including/neglecting certain things in the scheme changed the simulated system. By the time I got to the end, it seemed like a huge amount of information being provided to the reader but without much physical insight. I think the introduction would benefit from reducing the number of references where it is just stated that "X changes Y", and instead focus on a more limited number of studies and provide some physical pathways for how Y is changed by X. Otherwise, including all of these references isn't very informative; it just shows that changes to microphysics changed the simulations without any substance as to how or why.
- Fig. 10 and associated discussion on the impacts of microphysics on vertical velocity: These differences in vertical velocity seem really insignificant to me. The only real shift

you're talking about is in the 0.5-1 m/s bin, and the difference in the frequency of occurrence is less than 1%. Sure, it makes sense that less graupel in the profile may weaken the drag from condensate loading or perhaps have an effect like you described from Adams-Selin et al. (2013), but Fig. 10 is not convincing at all that these differences are not just noise. In fact, Fig. 10a shows very small but actually weaker vertical velocities in the PD scheme for the higher vertical velocity thresholds. I just don't think this effect is substantial enough to attribute the dynamical shift to microphysics as opposed to just perturbed system evolution. One could run a test by doing a small ensemble of PD runs with white noise added to the initial conditions to see if this very small shift is robust. But ultimately I don't think this is necessary, because I don't think this is an important result from your study and that there are more interesting things that you've already focused on. Personally, I think the manuscript would be improved by removing the discussion of impacts on vertical velocity and focusing instead on the more certain points. After all, these cases are synoptic lows with orographic enhancement, right? I wouldn't expect to see significant impacts on this type of system anyway compared to deep convection cases where cold pools are important for system evolution and where vertical motions are driven by buoyancy instead of synoptic-scale circulations.

**Specific Comments**
- Lines 72-76: This association between predicted vs. fixed particle density and the CCN concentrations is not very clear. You state that graupel density matters for appropriately simulating the impacts of varying CCN, but provide no details on the pathway for which this occurs. It would be helpful to the reader to briefly provide a clearer connection between the two rather than just saying one thing changed another–a physical explanation is prudent here.
- Lines 82-83: This sentence in particular is not very informative. Instead of saying that the simulated precipitation is simply sensitive to graupel density, tell us *how* it's sensitive to it. What happened to simulated precipitation when graupel density increased/decreased in Li et al. (2019)?
- At the beginning of Section 2, I'd recommend providing a brief few sentences on the background of the WDM6 scheme. For example, you don't mention what the 6 prognostic species are, but instead just start discussing the densities of the various species on Line 105. They should be cloud water, rain, cloud ice, snow, graupel, and CCN, correct? Related to this, you mention the "4 categories of ice" on Line 115. This can be very confusing because it seems you are referring to the species of ice in the WDM6 scheme, for which there are only 3. I assume you mean the 4 coefficients used to represent varying properties of the graupel species in the V-D and A-D relationships (as you state on Line 134)? This is not clear at this point in the manuscript. Recommend clearing this up where you first introduce it (Line 115). It could be helpful, though not necessary, to provide a table of the 6 species and their relevant m-D, V-D, and density parameters. This could also be a short Appendix addition. While the scheme is well-documented in past literature, a self-contained description often seems appropriate in a paper where only one scheme is being considered in such detail.

- I'm wondering how variable graupel density (e.g., as low as 100 kg/m3 as displayed in Fig. 1) impacts snow and the transition between these categories? Does the snow species rime? If so, how much before it is considered graupel? Related to this, Eq. 2, which is the source/sink processes for *graupel,* includes terms listed in Table 1 that have nothing to do with graupel, such as accretion of rain by snow, or snow by rain. So, is accretion of rain by snow a source term in that the mass from the snow + rain accretion is transferred to the graupel species?
- Lines 277-278: I'm not sure Fig. 6c actually shows that particles grow mostly via vapor deposition. The color-scale for deposition/sublimation uses the same contour thresholding as freezing and accretion, both of which appear to reach maximum values of 10 g/kg/s surrounding the core, and values of 0.01 enclosing most of the graupel mass. Ultimately this figure does not really show a closed budget. Although you refer to the DelaFrance et al. (2023) paper, I'm rather skeptical that vapor deposition is the primary growth process for graupel in a deep convection simulation--you'd have to prove to me that that isn't the case using a closed budget to accept this statement as true–for example by summing the individual process rates along horizontal levels and showing a profile. You state again on Line 288 that DEP is the main process producing graupel. Again, the color-scale/contouring doesn't really support this besides at the far reaches of the anvil. And even then, I wouldn't say that DEP is producing graupel in the anvil, but rather is just the most active process *growing* graupel in the anvil region, which would make sense. Deposition is certainly active, but (1) I doubt it's the primary *production* mechanism of graupel and (2) I'm skeptical it is the primary *growth* mechanism besides in the anvil region. To address (2), you'd have to show me and the readers a closed budget.
- Line 307: Is the total amount of surface snow actually reduced from WDM6_PD to WDM6_FD in Fig. 7c? This isn't clear from the map, where it looks like positive and negative values could offset each other in total. Recommend being more quantitative with this statement. While it is obvious that graupel is reduced in Fig. 7d, it could also be helpful to be more quantitative, perhaps by using a domain-accumulated snow/graupel relative difference and mentioning it in the text.
- Fig. 8: This isn't necessary, but the interpretation of this (and other) figure(s) would probably benefit by showing the 0 deg C level with a horizontal line.
- Line 343: "resulting in an increase in the amount of surface graupel deposited"--this statement may be a little confusing. You go on to explain why this is the case (basically, smaller graupel → faster fallspeeds (relative to FD) → faster sedimentation → greater surface graupel accumulation but lower graupel mass in the profile), and this is an interesting result, but when this statement is presented, the reader doesn't yet know the association/reasoning. This could be a good opportunity to state something along the lines of: "if graupel mass is reduced on average in the profile when using predicted density (Fig. 8c,f), why does it lead to greater surface graupel accumulation (Fig. 7c)?"
- Lines 354-358: Related to the prior point, I think the reader would really benefit from a figure showing, perhaps, profiles with percentiles of the mass-weighted mean diameter rather than just giving domain-horizontal-and-vertical averages. This point really seems to be getting to the crux of the interpretation, which is pretty interesting and deserves to

be highlighted. For example, Fig. 1 shows clearly that for all densities, the graupel terminal fall velocity with predicted graupel density is faster than with fixed density–but this is only unanimously true for particles smaller than ~ 1-2 mm, where you say your mean Dm lies. So I think this point is deserving of a little more attention. Of course this isn't necessary, but I think it would improve the manuscript.

- Fig. 9b and Line 360: I'm not really sure what you mean by "falling graupel mixing ratios depending on the mass-weighted terminal velocity" or by "the maximum level of falling graupel"; and I'm not really sure what's being shown in Fig. 9b,e. The units on the x-axis imply it is a mixing ratio, but there are negative values in the profile. Please revise the description of what you're showing here, because it makes the discussion around Line 360 rather hard to follow.

- Lines 362-364: While Fig. 8 clearly shows more snow mass in the profile on average, Fig. 9c,f doesn't show convincing evidence of greater snow deposition between the two simulations. I mean I see what you're talking about, but those differences seem remarkably small and insignificant relative to noise. Furthermore, for Fig. 9c,f, I would label these as "process rates" and not "production rates". Deposition is likely not *producing* graupel–and sublimation can't produce anything since it's a sink process.

- Lines 379-382: This is an important and interesting point! I'd love to see this highlighted more.

- Lines 397-406: This is an interesting result! Really shows the utility of what you've done here, which is great.

- Lines 450-451: Again, I don't think there is convincing evidence of a reduction in the strength of upward motion.
  Line 456: "but also predicts a wider range of fall velocity compared to the observed values"--isn't this the opposite of what is stated on Line 401, where you state "shows a much lower range of graupel fall velocity than the observed value"? And since Fig. 11's y-axis is logarithmic, isn't the range of fall velocities for the FD scheme smaller than observed (as stated on Line 401)?

- Lines 459-460: This sentence seems a little abrupt and out-of-place, but I think it deserves a little more attention and discussion. These last two sentences truly are a unique and interesting part of this study, but it's just mentioned in passing at the very end. It's not necessary, but expanding on this a little bit, and perhaps providing suggestions for a path forward to refine the simulated fall velocity, would be a worthy addition to guide future projects.

- This is picky semantics, but perhaps consider changing uses of "prognostic graupel density" to "predicted graupel density". The density is being derived from two prognostic variables, but the density itself is not prognostic.

**Technical Comments**

- Line 37 and others: Recommend changing the use of "convections" to the singular "convection"
- Line 42: change "cold pools" to "cold pool"
- Lines 42-45: You could probably combine these two sentences to just say that bow echoes *and* squall lines are sensitive to graupel fall speed parameters
- Line 49: "modelling" should be "modeling"

- Line 56: using "predicted" as second time in front of "rime density" is redundant. Consider removing second usage of the word.
- Lines 58 & 60: "ice-one" and "ice-two" doesn't really mean anything to the reader here, and don't appear to be necessary since you're just providing an example. Consider removing these names in parentheses
- Line 103: "$S_{BG}$ comprise" should be "$S_{BG}$ comprises"
- Line 103: $q_G$ seems arbitrarily placed here. One would assume it's the mass mixing ratio but this is not defined and comes after you talk about source/sink processes and before density. Please edit to make this sentence more clear.
- Line 104: It is customary to place the equation directly after mentioning it–otherwise the reader has to look ahead and then go back to read whatever description you've provided. Recommend putting Eq. 2 directly after its first reference and then explaining terms/variables after the equation has been introduced. Same thing for Equation 3.
- Line 107: You say $\rho_G$ can be prognosed, but it's actually $q_G$ and $B_G$ being prognosed. Recommend changing this to "$\rho_G$ can be predicted".
- Line 114: I don't really see a point to put a "G" subscript on the diameter (D) variable, since diameter is independent of species and these use gamma distributions anyway.
- Lines 114-115: The way this is stated is a bit confusing without specifically stating that you are referring to the original scheme. Recommend saying that "Further, $c_G$ is treated as a constant in the original scheme since…"
- Line 118: Again, recommend listing Equation 5 right after it is introduced, and then introduce Equations 6 and 7 with the explanations provided after.
- Line 127: Again recommending providing Eq. 9 directly after it is introduced.
- Line 135: This sentence is a bit confusing because you are saying the density of graupel is "assigned" in ranges in the modified scheme rather than being predicted via Eq. 3. Do you mean that the coefficients in the V-D relationship are derived for a given graupel density range, with the ranges given in Table 2? Please clear this up.
- Caption of Figure 1. You reference "Table 1" in regard to the "a" and "b" values, but these are in Table 2.
- Caption of Fig. 6: You say values are in units of mm. I think this should be m/s.
- Line 174: You say no case was selected for the air-sea interaction category, but you do list this as Case 7 in Table 3, so I don't understand what this sentence means.
- Line 219: "to 5 km grid" should be "to a 5 km grid"
- Line 287: "relative lower" should be "relatively lower"
- Line 288: "transported into anvil cloud region" should be "transported into the anvil cloud region"
- Caption of Fig. 10: Need to include "wind" after "positive vertical component"
- Line 304: Do you mean simulated *mass mixing* ratios? I would also refer to which panel of Fig. 7 you are talking about here, because it's not clear to me that the two schemes produce similar snow (c,g) and graupel (d,h) for the CL case (c,d). In fact, it seems that the differences between WDM6_FD and WDM6_PD in general are *larger* for the CL case compared to the WL case.
- Line 316: "between two experiments" should be "between the two experiments"
- Line 330: I would use *mass* mixing ratios–mixing ratio alone doesn't tell us much

- Line 351: I would say that the cells develop more extensively *in the vertical* here.
- Line 361: "As graupel fall quickly" should be "As graupel falls quickly"
- Line 362: Again, I'd be careful here to say it's suppression of graupel *generation*. Sure, less graupel mass in general throughout the profile would lead to less deposition and sublimation, but I'm not sure it's fair to say that weaker deposition suppresses graupel generation, but rather that it suppresses graupel growth.
- Line 365: "fall from a" should be "falls from a"
- Line 399: "rage" should be "range"
- LIne 431: "evolutions" should be "evolution"
- Lines 446-447: I think it would be more appropriate to say that "the change in surface precipitation is mainly attributed to the changes in surface snow"

---

## Referee Comment (RC2)

*General comments*

This paper explored the effect of prognostic graupel density in WDM6. The goal of this modification was to better model snowfall events with more realistic fall speed-diameter and mass-diameter relationships. It was interesting to see assumptions in existing methods get replaced by theories in new studies with reasonable success (lower RMSE). However, the biggest concern I have with the comparison between WDM6_PD with WDM6_FD is that fall speed-diameter relationship does not converge when they have the same density (Figure 1). I understand that it might have been implemented this way because the authors tried to keep the off-the-shelf fixed density model unchanged, but this implementation fails to facilitate a fair comparison between fixed vs. prognostic density because no one knows if the difference in simulation results come from the prognostic density or the vastly different parameters ($a_G$, $b_G$). It also leads to physical inconsistency: starting L350 (also Figure 9), the graupel in WDM6_PD falls faster than WDM6_FD despite its lower prognostic density (250-350 kg/m3 vs 500 kg/m3 in WDM6_FD) and smaller size. I recommend adding a modified WDM6_FD that is simply WDM6_PD when $\rho = 500 \ kg/m^3$ for a more meaningful comparison. This problem needs to be revisited because it could change the statistical skill scores shown in Table 4, but overall this is a study worth publishing once this problem is resolved.

*Specific comments*

L138. Why are the parameters in Table 2 so far off from the ones in the original WDM6 scheme? I don't think you need an extensive explanation but a one-sentence summary would be nice as it ties to points I made in the general comments.

L258/Fig. 5.

L316. Briefly explain equitable threat score (ETS) and why it's worth mentioning (even though the scores are similar).

L350. Physical inconsistency as mentioned in the general comments.

L379. The enhanced graupel fall velocity should not be attributed to the prognostic graupel density but rather the vastly different parameters ($a_G$, $b_G$) used.

L391. The slight enhancement of vertical velocity in the range of 0.1-0.5 m/s seems about equally insignificant for both CL and WL. The authors might also want to reexamine this figure with the WDM6_FD with modified parameters.

---

## Author Comment (AC1)

**Reviewer #1**

**Overview**

**This study describes the introduction of prognostic graupel bulk volume (and thus predicted graupel density) into a double-moment bulk microphysics scheme (WDM6) and evaluates its impact on (1) a 2D idealized squall line simulation and (2) simulations of several observed cases during a field campaign over the Korean Peninsula. The latter simulations are compared with observations including a large spatial array of surface meteorological stations that measured precipitation, a 2D video disdrometer, and a multi-angle snowflake camera. The study found that introducing predicted graupel density improved the representation of surface precipitation spatially and reduced statistical errors in most of the case study simulations, and that the new scheme reasonably represented the observed relationship between graupel density and fall velocity. While this study is founded on sound science questions and a robust methodology, and provides some very interesting results, there are several aspects that need to be addressed and improved before being considered for publication. Overall, the manuscript and interpretation of results could be improved by a more thorough description of the microphysics scheme as opposed to just the description of the new implementation. Since the study is heavily focused on a microphysical evaluation, there is justification to provide a little more background on the scheme structure. This study obviously involved a significant amount of work, but I am skeptical of some of the physical interpretations of the results. Apart from that, there are important and interesting results that are presented, and I think the manuscript would be particularly improved by focusing more closely on those robust results instead of casting such a wide net on the evaluation. The implementation of the new scheme and the vast constraints used to evaluate it against the observations are a huge undertaking that was performed well, and I think focusing on the observation-model comparison more would better highlight the novelty and success of the science that was performed.**

: Thank you for your valuable review and constructive feedback. Our manuscript has been revised including a more detailed description of the WDM6 microphysics scheme to offer better understanding of results. Additionally, we focused more closely on the observation-model comparisons by highlighting the verification of model results with 2DVD. We are confident that these changes make our study more accessible and informative for readers, better focusing the importance and effectiveness of the new predicted graupel density implementation in the WDM6 microphysics scheme.

**General/Major Comments**

**1. Overall, the description of the scheme needs to be revised/revisited. While some existence of knowledge should be assumed by the reader, the paper would benefit greatly from some additional information–even just a few basic sentences on the foundation of the WDM6 scheme. In addition, an improved description of the implemented, modified graupel species is needed, in particular how this implementation affects (or doesn't affect) the other ice species in the scheme.**

: In response to your comments, we have added a detailed introductory appendix that outlines the foundational information of the WDM6 scheme. This section provides the necessary background on the scheme's core principles, including the parameters that define the different categories.

Additionally, we have included the following sentences to explain how the modified graupel species affect the other ice species:

Line 150: "Therefore, the modified WDM6 incorporates varying $a_G$ and $b_G$ parameters in the $V_G$–D relationship and $c_G$ in the $M_G$–D relationship by implementing predicted graupel density. The several microphysics processes in the WDM6 can be affected by the newly derived $V_G$–D and $M_G$–D relationships. The microphysical processes of Pgmlt, Pgacw, Pgdep, Pgevp, and Ngacw are affected by $a_G$ and $b_G$ in the $V_G$–D relationship and Pgmlt, Pgaci, Pgacr, Pgdep, Pgevp, Pgacw, Ngaci, Ngacr, Ngeml, and Ngacw are

affected by $c_G$ in the $M_G$–D relationship. Since these processes act as source/sink for both mass mixing ratio and number concentration of cloud water, rain, cloud ice, snow, and graupel (Fig. A1 in the appendix), varying parameters with predicted graupel density can affect the mass mixing ratio and number concentration of liquid-phase hydrometeors as well as solid-phase hydrometeors."

**2. In general, much of the introduction was characterized by referencing past studies saying that including/neglecting certain things in the scheme changed the simulated system. By the time I got to the end, it seemed like a huge amount of information being provided to the reader but without much physical insight. I think the introduction would benefit from reducing the number of references where it is just stated that "X changes Y", and instead focus on a more limited number of studies and provide some physical pathways for how Y is changed by X. Otherwise, including all of these references isn't very informative; it just shows that changes to microphysics changed the simulations without any substance as to how or why.**

: Thank you for your suggestion. In response to your suggestion, I have undertaken a substantial revision of the introduction, detailing the physical pathways for how Y is changed by X and removing the references that were published a long time ago. The following sentence has been modified in the revised manuscript:

Line 38: "Morrison and Milbrandt (2011) demonstrated that different approaches in treating graupel or hail produce distinct differences in storm structure, precipitation, and cold pools strength for idealized supercells. This is because graupel leads to more anvil condensate and weaker cold pools compared to hail. Bryan and Morrison (2012) showed that the fall velocities of graupel and hail affect the simulated reflectivity and dynamics for an idealized squall line. Simulations with graupel instead of hail produce convective regions that are too wide and have lower reflectivity, primarily due to the slower fall velocity of graupel compared to hail. Adams-Selin et al. (2013) reported that the development of a bow echo is highly sensitive to the parameters defining the fall velocities of graupel and hail. The simulations with slower-falling graupel-like particle created a wider stratiform region and stronger cold pool, allowing for more melting and evaporation, which helped generating bowing segments earlier than in the faster-falling hail-like simulations."

Line 65: "Johnson et al. (2016) evaluated the reproducibility of the polarization signatures in supercell storms for several partially or fully two-moment (2M) schemes. Realistic signatures were obtained only with those microphysics schemes that predicted graupel density. Predicted graupel density assigns high-density frozen drops to the graupel category, resulting in relatively high-density graupel that can later grow into hail. These differences in the treatment of rimed-ice processes allow hail to grow larger and produce a much more prominent hail signature.

Line 73: "Since CCN concentration affects cloud droplet number concentration and mean droplet diameter, the model's microphysical response depends on how well parameterized processes involving the ice phase account for droplet size effects. Mean droplet size impacts graupel growth directly through the collection efficiency between graupel and droplets. Additionally, predicted graupel density influences graupel growth by increasing graupel fall speeds and enhancing accretion rates. Based on their analysis, they suggested that an accurate representation of graupel in microphysics schemes is crucial for appropriately simulating the effects of changes in the concentration of cloud condensation nuclei in selected systems."

**3. Fig. 10 and associated discussion on the impacts of microphysics on vertical velocity: These differences in vertical velocity seem really insignificant to me. The only real shift you're talking about is in the 0.5-1 m/s bin, and the difference in the frequency of occurrence is less than 1%. Sure, it makes sense that less graupel in the profile may weaken the drag from condensate loading or perhaps have an effect like you described from Adams-Selin et al. (2013), but Fig. 10 is not convincing at all that these differences are not just noise. In fact, Fig. 10a shows very small but actually weaker vertical velocities in the PD scheme for the higher vertical velocity thresholds. I just don't think this effect is substantial enough to attribute the dynamical shift to microphysics as opposed to just perturbed system evolution. One could run a test by doing a small ensemble of PD runs with white noise added to the initial conditions to see if this very small shift is robust. But ultimately I don't think this is necessary, because I don't think this is an important result from your study and that there are more interesting things that**

you've already focused on. Personally, I think the manuscript would be improved by removing the discussion of impacts on vertical velocity and focusing instead on the more certain points. After all, these cases are synoptic lows with orographic enhancement, right? I wouldn't expect to see significant impacts on this type of system anyway compared to deep convection cases where cold pools are important for system evolution and where vertical motions are driven by buoyancy instead of synoptic-scale circulations.

: We concur with your assessment that the differences in vertical velocity depicted in Fig. 10 are minimal and do not substantiate the suggested impacts of microphysical changes as robustly as required for a conclusive argument. Considering your comments and after careful consideration, Fig. 10 and its related descriptions are removed in the revised manuscript.

**Specific Comments**

**4. Lines 72-76: This association between predicted vs. fixed particle density and the CCN concentrations is not very clear. You state that graupel density matters for appropriately simulating the impacts of varying CCN, but provide no details on the pathway for which this occurs. It would be helpful to the reader to briefly provide a clearer connection between the two rather than just saying one thing changed another–a physical explanation is prudent here.**

: Thank you for your insightful feedback. To address your comment, we have provided a clearer explanation of the connection between predicted versus fixed particle density and the impacts of varying cloud condensation nuclei (CCN) concentrations as following:

Line 72: "~, particularly related to different number concentrations of cloud condensation nuclei (CCN) in a mid-latitude continental squall line."

Line 73: "Since CCN concentration affects cloud droplet number concentration and mean droplet diameter, the model's microphysical response depends on how well parameterized processes involving the ice phase account for droplet size effects. Mean droplet size impacts graupel growth directly through the collection efficiency between graupel and droplets. Additionally, predicted graupel density influences graupel growth by increasing graupel fall speeds and enhancing accretion rates. Based on their analysis, they suggested that an accurate representation of graupel in microphysics schemes is crucial for appropriately simulating the effects of changes in the concentration of cloud condensation nuclei in selected systems."

**5. Lines 82-83: This sentence in particular is not very informative. Instead of saying that the simulated precipitation is simply sensitive to graupel density, tell us how it's sensitive to it. What happened to simulated precipitation when graupel density increased/decreased in Li et al. (2019)?**

: In response to reviewer's comment, the following sentence has added in the revised manuscript:

Line 85: "Li et al. (2019) showed that the simulated precipitation exhibits significant sensitivity to changes in graupel density in the WDM6 scheme. Specifically, a lower-graupel density tends to contribute more to one-month precipitation amounts below 100 mm and less to those above 100 mm during the autumn season. Conversely, a higher-graupel density shows the opposite pattern."

**6. At the beginning of Section 2, I'd recommend providing a brief few sentences on the background of the WDM6 scheme. For example, you don't mention what the 6 prognostic species are, but instead just start discussing the densities of the various species on Line 105. They should be cloud water, rain, cloud ice, snow, graupel, and CCN, correct? Related to this, you mention the "4 categories of ice" on Line 115. This can be very confusing because it seems you are referring to the species of ice in the WDM6 scheme, for which there are only 3. I assume you mean the 4 coefficients used to represent varying properties of the graupel species in the V-D and A-D relationships (as you state on Line 134)? This is not clear at this point in the manuscript. Recommend clearing this up where you first introduce it (Line 115). It could be helpful, though not necessary, to provide a table of the 6 species and their relevant m-D, V-D, and density parameters. This could also be a short Appendix addition. While the scheme is well-**

**documented in past literature, a self-contained description often seems appropriate in a paper where only one scheme is being considered in such detail.**

: Thank you for your valuable feedback regarding the introduction of the WDM6 scheme in Section 2. As you suggested, we have revised the beginning of this section to include a description of the hydrometeor characteristics of the WDM6 scheme. We have also included a table in Appendix, detailing the parameters of hydrometeors characteristics as following:

**Table A1.** Parameters for hydrometeor (rain, ice, snow and graupel) characteristics in WDM6 scheme.

| | $V_x - D_x$ relationship | | $M_x - D_x$ relationship | | Shape parameter ($\mu_x$) | Density ($\rho_x$) |
|---|---|---|---|---|---|---|
| | $a_x$ | $b_x$ | $c_x$ | $d_x$ | | |
| Rain | 841.9 | 0.8 | $\frac{\pi\rho_R}{6}$ | 3 | 1 | 1000 |
| Cloud ice | $2.71 \times 10^3$ | 1.0 | $\frac{\pi\rho_I}{6}$ | 3.0 | 0 | 500 |
| Snow | 11.72 | 0.41 | $\frac{\pi\rho_S}{6}$ | 3.0 | 0 | 100 |
| Graupel | 330.0 | 0.8 | $\frac{\pi\rho_G}{6}$ | 3.0 | 0 | 500 |

Line 101: "In the original WDM6 scheme,  characteristics are pre-defined using the static value of density ($\rho_G$), constant coefficients for the mass ($\underline{M_G}$)–diameter ($\underline{D_G}$) and fall velocity ($\underline{V_G}$)–$\underline{D_G}$ relationships." → "In the original WDM6 scheme, characteristics of hydrometeors are pre-defined using the static value of density ($\rho_X$), constant coefficients for the mass ($\underline{M_X}$)–diameter ($\underline{D}$) and fall velocity ($\underline{V_X}$)–$\underline{D}$ relationships. Here, X represent the species of hydrometeors including cloud water, rain, cloud ice, snow, and graupel. The specific values of parameters are available in Table A1 of the Appendix."

Thank you for highlighting the potential confusion regarding the phrase, "four categories of ice particles" which is initially described for the study of Mitchell (1996). To make the meaning of sentence more clearly, the corresponding sentences are revised as following:

Line 158: "Meanwhile, Mitchell (1996) addressed that the Reynolds number ($Re$)–Best number ($X$) relationship produces the power-law expressions of fall velocity according to ice particle types based on the relationships of mass and projected area with the dimensions as Eq. (5)."

We recognize that the use of the four coefficients to represent varying properties of the graupel species in the M-D and A-D relationships in equation (10) and (11) was not clearly explained in the original manuscript. To address this, we have revised the sentence as follows:

Line 126: "In our modified WDM6, $c_G$ varies with the predicted $\rho_G$ (Eq. (3)). The coefficients of the area ($A_G$)–D relationship ($A_G = \gamma D^\sigma$), $\gamma$ and $\sigma$, are set to $\frac{\pi}{4}$ and 2.0, respectively due to the sphere-shaped graupel in the WDM6 scheme."

Line 146: "Further, $c_G$ and $d_G$ represent the coefficients of the $M_G$–D relationship, while $\gamma$ and $\sigma$ are the coefficients of the $A_G$–D relationship,  respectively. ."

**7. I'm wondering how variable graupel density (e.g., as low as 100 kg/m3 as displayed in Fig. 1) impacts snow and the transition between these categories? Does the snow species rime? If so, how much before it is considered graupel? Related to this, Eq. 2, which is the source/sink processes for graupel, includes terms listed in Table 1 that have nothing to do with graupel, such as accretion of rain by snow, or snow by rain. So, is accretion of rain by snow a source term in that the mass from the snow + rain accretion is transferred to the graupel species?**

: Thank you for your comment. As we mentioned in comment #1, we have included the following sentences to explain how the modified graupel species affect the other ice species:

Line 155: "The several microphysics processes in the WDM6 can be affected by the newly derived $V_G$–D and $M_G$–D relationships. The microphysical processes of Pgmlt, Pgacw, Pgdep, Pgevp, and Ngacw are affected by $a_G$ and $b_G$ in the $V_G$–D relationship and Pgmlt, Pgaci, Pgacr, Pgdep, Pgevp, Pgacw, Ngaci, Ngacr, Ngeml, and Ngacw are affected by $c_G$ in the $M_G$–D relationship. Since these processes act as source/sink for both mass mixing ratio and number concentration of cloud water, rain, cloud ice, snow, and graupel (Fig. A1 in the appendix), varying parameters with predicted graupel density can affect the mass mixing ratio and number concentration of liquid-phase hydrometeors as well as solid-phase hydrometeors."

To provide a clearer description of snow as suggested, we have added the following sentence to the revised manuscript:

Line 104: "In the WDM6 scheme, 'snow' is defined as an unrimed ice phase (large crystals-aggregates) with a standard density of 100 kg m$^{-3}$, indicating that it does not undergo riming. Conversely, 'graupel' is characterized as heavily rimed crystal particles that have not undergone wet growth. In nature, graupel has a wide range of densities according to the degree of riming. However, the original WDM6 scheme is unable to simulate this variability in graupel density as it undergoes riming because it uses a predefined constant value for graupel density."

Regarding the accretion processes mentioned in Eq. 2 and Table 1, these processes, including the accretion of rain by snow, indeed contribute to the formation and growth of graupel (or snow). To help in understanding the source/sink flow chart of these microphysical processes, we have added the following sentences with Figure A1 to Appendix:

Line 557: "One of the accretion processes, Psacr, represents the accretion between snow and rain particles, which primarily contributes to the formation of graupel or snow. When the mass mixing ratios of both rain and snow are greater (smaller) than 1.e-4 kg kg$^{-1}$, it contributes to the formation of graupel (snow). This process acts as a source process for the graupel or snow mass mixing ratio and as a sink process for the rain mass mixing ratio (Fig. A1a)"

[Figure]

**Figure A1.** Flowcharts of microphysical processes for predicting (a) mass mixing ratio ($S_{q_x}$) and (b) number concentration ($S_{N_x}$) of hydrometeors in WDM6 scheme. The number concentrations of hydrometeors in the green boxes are predicted only (e.g., cloud water, cloud ice, rain, and cloud condensation nuclei (CCN)). Microphysical terms drawn with red (blue) are activated when the temperature is above (below) 0°C. Terms drawn in black are activated regardless of temperature.

**8. Lines 277-278: I'm not sure Fig. 6c actually shows that particles grow mostly via vapor deposition. The color-scale for deposition/sublimation uses the same contour thresholding as freezing and accretion, both of which appear to reach maximum values of 10 g/kg/s surrounding the core, and values of 0.01 enclosing most of the graupel mass. Ultimately this figure does not really show a closed budget. Although you refer to the DelaFrance et al. (2023) paper, I'm rather skeptical that vapor deposition is the primary growth process for graupel in a deep convection simulation--you'd have to prove to me that that isn't the case using a closed budget to accept this statement as true–for example by summing the individual process rates along horizontal levels and showing a profile. You state again on Line 288 that DEP is the main process producing graupel. Again, the color-scale/contouring doesn't really support this besides at the far reaches of the anvil. And even then, I wouldn't say that DEP is producing graupel in the anvil, but rather is just the most active process growing graupel in the anvil region, which would make sense. Deposition is certainly active, but (1) I doubt it's the primary production mechanism of graupel and (2) I'm skeptical it is the primary growth mechanism besides in the anvil region. To address (2), you'd have to show me and the readers**

: Thank you for your comment. In response to the reviewer's comment, we have included Figure R1 to show vertical profiles for the time-domain averaged source/sinks of the graupel mass mixing ratio. Our analysis of the process rate in Figure R1 indicates that ACC is the most effective sources of graupel, while the magnitude of DEP is relatively smaller than that of ACC at 1, 2, and 4 hour. To clarify the microphysical budget analysis, we have deleted the following sentences in the revised manuscript:

"The microphysical budget analysis shows that particles mostly grow by vapor deposition in the initial stage (Fig. 6c). The sensitivities in ice-phase particle growth and transport due to variabilities in the riming processes over an orographic barrier were examined by using a unique Lagrangian particle-based precipitation model in the study by DeLaFrance et al. (2023). This study revealed that particles initially grow by deposition and have a lower effective density. Very dense graupel ($\rho_G$ values of 900 kg m$^{-3}$) are located in the marginal regions of updraft cores (Fig. 6b)."

Also, we have added Figure R1 to the supplementary figures, the following sentence has added in the revised manuscript:

Line 323: "The vertical profiles for the domain-averaged major source/sink microphysics processes are presented in Figure S1 of the Supplement. ACC and MLT are analyzed as the most active source and sink processes in both WDM6_PD and WDM6_FD"

Additionally following sentence has been revised from "Over the corresponding region, DEP is the main process producing graupel." to "Over the corresponding region, DEP and ACC are the primary active processes for growing graupel."

[Figure]

**Figure R1.** Vertical profiles for the domain-averaged (a) sources and (b) sinks of graupel mass mixing ratio in WDM6_FD (Solid line) and WDM6_PD (dashed line) at 1 hour (a and b), 2 hour (c and d), and 4 hour (e and f). The main source processes, namely, deposition (Pgdep; DEP), accretion (mean of Paacw, Psacr and Pgacr; ACC) and freezing (Pgfrz; FRZ) are plotted with the major sink processes, namely, sublimation (Pgsub; SUB) and melting (Pgmlt; MLT).

**9. Line 307: Is the total amount of surface snow actually reduced from WDM6_PD to WDM6_FD in Fig. 7c? This isn't clear from the map, where it looks like positive and negative values could offset each other in total. Recommend being more quantitative with this statement. While it is obvious that graupel**

**is reduced in Fig. 7d, it could also be helpful to be more quantitative, perhaps by using a domain-accumulated snow/graupel relative difference and mentioning it in the text.**

: To provide a clearer and more quantitative comparison as suggested, we have added the following sentence associated with Figure 8 in the revised manuscript:

Line 346: "Specifically, the total surface snow is reduced by 93% (domain-averaged snow amount is 0.75 mm in WDM6_FD and 0.80 mm in WDM6_PD), and surface graupel shows an increase of 124% (domain-averaged graupel amount is 0.64 mm in WDM6_FD and 0.51 mm in WDM6_PD) in WDM6_PD compared to WDM6_FD."

Line 353: "Surface snow decreases significantly by 92% in WDM6_PD (domain-averaged snow amount is 0.77 mm in WDM6_FD and 0.84 mm in WDM6_PD), compared to WDM6_FD, while the surface graupel increases by 121 % (domain-averaged graupel amount is 0.21 mm in WDM6_FD and 0.18 mm in. WDM6_PD) (Figs. 8g and h)."

**10. Fig. 8: This isn't necessary, but the interpretation of this (and other) figure(s) would probably benefit by showing the 0 deg C level with a horizontal line.**

: We agree that the inclusion of the 0°C level as a horizontal line would significantly enhance the interpretability of this and other figures. As such, we have redrawn Figure 9 to include a horizontal line indicating the 0°C level and modified its caption.

Line 394: "The sum of snow and graupel mass mixing ratios (g kg$^{-1}$) is indicated by red lines, and the 0°C level by the grey dashed horizontal line."

**11. Line 343: "resulting in an increase in the amount of surface graupel deposited"—this statement may be a little confusing. You go on to explain why this is the case (basically, smaller graupel → faster fallspeeds (relative to FD) → faster sedimentation → greater surface graupel accumulation but lower graupel mass in the profile), and this is an interesting result, but when this statement is presented, the reader doesn't yet know the association/reasoning. This could be a good opportunity to state something along the lines of: "if graupel mass is reduced on average in the profile when using predicted density (Fig. 8c,f), why does it lead to greater surface graupel accumulation (Fig. 7c)?"**

: In response to the reviewer's comment, we have added the following sentence to include a more explicit discussion of the underlying processes before presenting the association/reasoning of increased surface graupel accumulation.

Line 387: "~, resulting in an increase in the amount of surface graupel deposited (Figs. 7d and h)." → "~, resulting in an increase in the amount of surface graupel deposited (Figs. 8d and h). The reason for the lower graupel mass (Figs. 9c and f), despite the greater surface graupel accumulation (Figs. 8d and h) in WDM6_PD, will be analyzed in the subsequent Figures 10 and 11."

**12. Lines 354-358: Related to the prior point, I think the reader would really benefit from a figure showing, perhaps, profiles with percentiles of the mass-weighted mean diameter rather than just giving domain-horizontal-and-vertical averages. This point really seems to be getting to the crux of the interpretation, which is pretty interesting and deserves to be highlighted. For example, Fig. 1 shows clearly that for all densities, the graupel terminal fall velocity with predicted graupel density is faster than with fixed density–but this is only unanimously true for particles smaller than ~ 1-2 mm, where you say your mean Dm lies. So I think this point is deserving of a little more attention. Of course this isn't necessary, but I think it would improve the manuscript.**

:. In response to the reviewer's suggestion, we have added Figures 10b and d, which presents profiles of $D_m$ with percentiles. Additionally, we have added the following sentences in the revised manuscript:

[Figure]

**Figure 20.** Vertical profiles for the time-domain-averaged $\rho_G$ (kg m$^{-3}$) for (a) CL and (c) WL cases with WDM6_PD. Time-domain-averaged mass-weighted mean diameter ($D_m$) (mm) with WDM6_PD and WDM6_FD are drawn in (b) and (d) for CL and WL cases. The solid and dashed lines represent WDM6_FD and WDM6_PD, respectively.

Line 401: "The time-domain-averaged mass-weighted mean diameter ($D_m$) in WDM6_PD is greatly reduced compared to WDM6_FD (Figs. 10b and d). In the CL case, the range of $D_m$ is quite wider below the 4-km level, indicating more variability in graupel sizes than the WL case. In both cases, WDM6_PD presents smaller graupel than WDM6_FD, especially over the lower level."

Line 404: "In WDM6_PD, the time-domain-averaged mass-weighted mean diameter ($D_m$) is simulated as 0.110 and 0.191 mm for the CL and WL cases, respectively, whereas in WDM6_FD, it is simulated as 0.133 (CL) and 0.199 (WL) mm, indicating that WDM6_PD simulates smaller graupel diameters." → In WDM6_PD, the time-domain and vertical-averaged $D_m$ is simulated as 0.110 mm and 0.191 mm for the CL and WL cases, respectively, whereas in WDM6_FD, it is simulated as 0.133 mm (CL) and 0.199 mm (WL)."

Line 410: "In the CL case, WDM6_PD simulates $\rho_G$ with a maximum value of 220 kg m$^{-3}$ and $D_m$ with a maximum value of 0.44 mm at around the 2 km level (Figs. 10a and b)."

Line 419: "In the WL case, graupel, which exists up to the 10 km level, $\rho_G$ increases significantly up to a value of 350 kg m$^{-3}$ at 1 km level (Fig. 10c). Even though $D_m$ of WDM6_PD is larger than that of WDM6_FD above the 3-km level, graupel particles in WDM6_PD have a greater falling velocity (Figs.10d and 1) and fall from a relatively higher level of 8 km compared to WDM6_FD (Fig. 11c)."

**13. Fig. 9b and Line 360: I'm not really sure what you mean by "falling graupel mixing ratios depending on the mass-weighted terminal velocity" or by "the maximum level of falling graupel"; and I'm not really sure what's being shown in Fig. 9b,e. The units on the x-axis imply it is a mixing ratio, but there are negative values in the profile. Please revise the description of what you're showing here, because it makes the discussion around Line 360 rather hard to follow.**

: To make the meaning of the x-axis label of Figures 11a and c clearly, it has been revised to "$q_z$-$q_{z-1}$ (g kg$^{-1}$)" in the revised manuscript. Additionally, to clarify the meaning of figures, we have revised the caption from "Time-domain averaged falling graupel mass mixing ratios depending on the mass-weighted terminal velocity" to "Time-domain averaged difference in graupel mass mixing ratio between the levels 'z' ($q_z$) and 'z-1' ($q_{z-1}$) due to sedimentation" in the revised manuscript.

**14. Lines 362-364: While Fig. 8 clearly shows more snow mass in the profile on average, Fig. 9c,f doesn't show convincing evidence of greater snow deposition between the two simulations. I mean I see what you're talking about, but those differences seem remarkably small and insignificant relative to noise. Furthermore, for Fig. 9c,f, I would label these as "process rates" and not "production rates". Deposition is likely not producing graupel–and sublimation can't produce anything since it's a sink process.**

: In response to reviewer's comment, we have revised the terminology as "process rates".

To illustrate the greater snow mass of WDM6_PD for the CL case (Fig. 9c), we have analyzed snow advection in Figure R2. It is evident that the inland area, receiving abundant precipitation, presents more snow advection at the 850 hPa level in WDM6_PD compared to WDM6_FD (Fig. R1b). The snow advection towards the inland area by the strengthened northeasterly wind can certainly enhance the snow mass mixing ratio in WDM6_PD.

Additionally, more efficient accretion between cloud water and snow/graupel (Paacw), due to the increased snow advection, further contributes to the increased snow mass mixing ratio in WDM6_PD (Fig. R1c). To provide a clearer reason for the increased snow mass in WDM6_PD, we have added Figure R2 to the supplementary figures, the following sentence has added in the revised manuscript:

Line 416: "Furthermore, the northeastern inland area, receiving abundant precipitation, exhibits more positive snow advection at the 850 hPa level in WDM6_PD compared to WDM6_FD (Fig. S6 in the Supplement). Increased snow advection towards the inland area enhances the snow mass mixing ratio in WDM6_PD. Additionally, efficient Paacw with more available snow mass can contribute to the increased snow mass mixing ratio in WDM6_PD."

[Figure]

**Figure R3.** Snow advection (g kg$^{-1}$ s$^{-1}$) and wind vector (m s$^{-1}$) at 850hPa for CL case from (a) WDM6_FD, and (b) the difference between WDM6_PD and WDM6_FD (WDM6_PD minus WDM6_FD). The vertical profiles of the time-domain-averaged Paacw process for C1 case are shown in (c). The solid (dashed) line represents WDM6_FD (WDM6_PD).

**15. Lines 379-382: This is an important and interesting point! I'd love to see this highlighted more.**

: Thank you for recognizing the significance of verifying our simulations with the 2DVD data. To highlight our verification results further, we have added the following sentence in the revised manuscript:

Line 450: "…Although WDM6_PD simulates larger ranges of fall velocity and lower ranges of $\rho_G$, it is closer to the observations than WDM6_FD. Our analysis hights that WDM6_PD with varying graupel density results in faster fall velocities, leading to more efficient sedimentation processes, which affect the spatial distribution and amount of graupel mass mixing ratio both in the atmosphere and on the surface. By predicting graupel density, WDM6_PD can produce more realistic characteristics of graupel particles, including their density and fall velocity."

**16. Lines 397-406: This is an interesting result! Really shows the utility of what you've done here, which is great. I'd love to see this highlighted more.**

: Please refer to the response for comment #15 above.

**17. Lines 450-451: Again, I don't think there is convincing evidence of a reduction in the strength of upward motion.**

: As previously mentioned in response to comment #3, we have removed Figure 10 and its related analysis in the revised manuscript.

**18. Line 456: "but also predicts a wider range of fall velocity compared to the observed values"--isn't this the opposite of what is stated on Line 401, where you state "shows a much lower range of graupel fall velocity than the observed value"? And since Fig. 11's y-axis is logarithmic, isn't the range of fall velocities for the FD scheme smaller than observed (as stated on Line 401)?**

: To address the reviewer's comment, we have modified the following sentence in the revised manuscript:

Line 501: "but also predicts a wider range of fall velocity compared to the observed values." → "but also predicts a lower range of fall velocity compared to the observed values."

**19. Lines 459-460: This sentence seems a little abrupt and out-of-place, but I think it deserves a little more attention and discussion. These last two sentences truly are a unique and interesting part of this study, but it's just mentioned in passing at the very end. It's not necessary, but expanding on this a little bit, and perhaps providing suggestions for a path forward to refine the simulated fall velocity, would be a worthy addition to guide future projects.**

: In response to the reviewer's feedback, we have expanded this discussion to provide potential avenues for further refinement of the simulated fall velocities. To this end, we have added the following sentences in the revised manuscript:

Line 504: "The $V_G$–D relationship in the modified WDM6 is derived using the least-squares method in a log–log space at the given graupel density; therefore, there is room to further refine the simulated fall velocity. The derived $V_G$–D relationship in our research could be refined by incorporating a broader range of graupel observational data, including hexagonal, conical, lump graupel, or graupel-like snow. Improvements in the representation of $V_G$–D relationship can lead to better simulation of precipitation and microphysical processes in environments where various types of graupel are generated. Additionally, the potential benefits of the predicted graupel density could be further evaluated in future works through comparison with additional observational data such as sonde and satellite.

**20. This is picky semantics, but perhaps consider changing uses of "prognostic graupel density" to "predicted graupel density". The density is being derived from two prognostic variables, but the density itself is not prognostic.**

: Thank you for your suggestion to change the term "prognostic graupel density" to "predicted graupel density" to better reflect the derivation process. We agree that this terminology more accurately describes the density as being derived from prognostic variables rather than being prognostic itself. In response to your feedback, we have updated the manuscript to replace all instances of "prognostic graupel density" with "predicted graupel density." throughout in revised manuscript.

Additionally, the title of this paper has been revised as "Introducing Graupel Density Prediction in Weather Research and Forecasting (WRF) Double-Moment 6-Class (WDM6) Microphysics and Evaluation of the Modified Scheme During the ICE-POP Field Campaign"

**Technical Comments**

**21.    Line 37 and others: Recommend changing the use of "convections" to the singular "convection"**

: Revised accordingly.

**22.    Line 42: change "cold pools" to "cold pool"**

: Revised accordingly.

**23.    Lines 42-45: You could probably combine these two sentences to just say that bow echoes and squall lines are sensitive to graupel fall speed parameters**

: We have explained each of the two references in detail in response to comment 2. Please refer to the response for comment #2 above.

**24.    Line 49: "modelling" should be "modeling"**

: Revised accordingly.

**25.    Line 56: using "predicted" as second time in front of "rime density" is redundant. Consider removing second usage of the word.**

: In response to your comment, we have modified the following sentence:

Line 57: "Morrison and Milbrandt (2015) later developed the Predicted Particle Properties (P3) bulk microphysics scheme that predicts the rime mass fraction, rime volume, and  rime density for a single generic ice-phase category."

**26.    Lines 58 & 60: "ice-one" and "ice-two" doesn't really mean anything to the reader here, and don't appear to be necessary since you're just providing an example. Consider removing these names in parentheses**

: Revised accordingly.

**27.    Line 103: "SBG comprise" should be "SBG comprises"**

: Revised accordingly.

**28.    Line 103: qG seems arbitrarily placed here. One would assume it's the mass mixing ratio but this is not defined and comes after you talk about source/sink processes and before density. Please edit to make this sentence more clear.**

: In response to your comment, we have modified the following sentence:

Line 114: " $S_{B_G}$ comprise several microphysical source/sink processes $q_G$ and density of specific hydrometeors ($\rho_X$) according to Eq. (2)." → " $S_{B_G}$ comprise several microphysical source/sink processes for mass mixing ratio of graupel ($q_G$) and density of specific hydrometeors ($\rho_X$), as defined in Eq. (2)."

**29.    Line 104: It is customary to place the equation directly after mentioning it–otherwise the reader has to look ahead and then go back to read whatever description you've provided. Recommend putting Eq. 2 directly after its first reference and then explaining terms/variables after the equation has been introduced. Same thing for Equation 3.**

: Revised accordingly.

**30. Line 107: You say ρG can be prognosed, but it's actually qG and BG being prognosed. Recommend changing this to "ρG can be predicted".**

: Revised accordingly.

**31. Line 114: I don't really see a point to put a "G" subscript on the diameter (D) variable, since diameter is independent of species and these use gamma distributions anyway.**

: Revised accordingly.

**32. Lines 114-115: The way this is stated is a bit confusing without specifically stating that you are referring to the original scheme. Recommend saying that "Further, cG is treated as a constant in the original scheme since…"**

: In response to your comment, we have modified the following sentence:

Line 125: "Further, $c_G$ is treated as a constant since $\rho_G$ is set as a constant (500 kg m$^{-3}$)" → "Further, $c_G$ is treated as a constant since $\rho_G$ in the original WDM6 scheme is set as a constant (500 kg m$^{-3}$)."

**33. Line 118: Again, recommend listing Equation 5 right after it is introduced, and then introduce Equations 6 and 7 with the explanations provided after.**

: Revised accordingly.

**34. Line 127: Again recommending providing Eq. 9 directly after it is introduced.**

: Revised accordingly.

**35. Line 135: This sentence is a bit confusing because you are saying the density of graupel is "assigned" in ranges in the modified scheme rather than being predicted via Eq. 3. Do you mean that the coefficients in the V-D relationship are derived for a given graupel density range, with the ranges given in Table 2? Please clear this up.**

: In response to your comment, we have modified the following sentence:

Line 148: " $a_G$ and $b_G$ in the $V_G$–D relationship are derived at the predicted $\rho_G$, which is in the range of 100–900 kg m$^{-3}$, at intervals of 100 kg m$^{-3}$ to facilitate the transition between aggregate and rime particles (Straka and Mansell, 2005), using the least-squares method in a log–log space over a range of $D_G$ of 0.3–20 mm (Table 2)."

**36. Caption of Figure 1. You reference "Table 1" in regard to the "a" and "b" values, but these are in Table 2.**

: Revised accordingly.

**37. Caption of Fig. 6: You say values are in units of mm. I think this should be m/s.**

: Revised accordingly.

**38. Line 174: You say no case was selected for the air-sea interaction category, but you do list this as Case 7 in Table 3, so I don't understand what this sentence means.**

: In response to your comment, we have modified the following sentence:

Line 196: "However, no case was selected for the air–sea interaction category because only one event from this category was identified during the ICE-POP field campaign." → "Although Case7 is listed in Table 3 as an air-sea interaction event, it is not selected for detailed analysis because only one event from this category was identified during the ICE-POP field campaign."

**39. Line 219: "to 5 km grid" should be "to a 5 km grid"**

: Revised accordingly.

**40.  Line 287: "relative lower" should be "relatively lower"**

: Revised accordingly.

**41.  Line 288: "transported into anvil cloud region" should be "transported into the anvil cloud region"**

: Revised accordingly.

**42.  Caption of Fig. 10: Need to include "wind" after "positive vertical component"**

: As previously mentioned in response to comment #3, we have removed Figure 10 in the revised manuscript.

**43.  Line 304: Do you mean simulated mass mixing ratios? I would also refer to which panel of Fig. 7 you are talking about here, because it's not clear to me that the two schemes produce similar snow (c,g) and graupel (d,h) for the CL case (c,d). In fact, it seems that the differences between WDM6_FD and WDM6_PD in general are larger for the CL case compared to the WL case.**

: To make the meaning of sentence clearly, we have modified the following sentence:

Line 341: "WDM6_FD and WDM6_PD provide similar simulated ratios of surface snow and graupel for the CL case." → "The surface snow amount is similar to the surface graupel one in both WDM6_FD and WDM6_PD for CL case."

**44.  Line 316: "between two experiments" should be "between the two experiments"**

: Revised accordingly.

**45.  Line 330: I would use mass mixing ratios–mixing ratio alone doesn't tell us much**

: In response to the reviewer's comment, we have modified 'mixing ratio' to 'mass mixing ratio' throughout the revised manuscript.

**46.  Line 351: I would say that the cells develop more extensively in the vertical here.**

: In response to your comment, we have modified the following sentence:

Line 398: "As shown in Fig. 8, convective cells develop more extensively in the WL case than in the CL case." → "As shown in Fig. 9, convective cells develop more extensively in the vertical direction in the WL case than in the CL case."

**47.  Line 361: "As graupel fall quickly" should be "As graupel falls quickly"**

: Revised accordingly.

**48.  Line 362: Again, I'd be careful here to say it's suppression of graupel generation. Sure, less graupel mass in general throughout the profile would lead to less deposition and sublimation, but I'm not sure it's fair to say that weaker deposition suppresses graupel generation, but rather that it suppresses graupel growth.**

: In response to your comment, we have modified the following sentence:

Line 412: "As graupel fall quickly in WDM6_PD, graupel deposition (Pgdep) decreases, leading to the suppression of graupel generation and sublimation (Pgsub) (Fig. 9c)." → "As graupel falls quickly in WDM6_PD, graupel deposition (Pgdep) decreases, leading to the suppression of graupel growth and sublimation (Pgsub) (Fig. 11b)."

**49.  Line 365: "fall from a" should be "falls from a"**

: Revised accordingly.

**50. Line 399: "rage" should be "range"**

: Revised accordingly.

**51. Line 431: "evolutions" should be "evolution"**

: Revised accordingly.

**52. Lines 446-447: I think it would be more appropriate to say that "the change in surface precipitation is mainly attributed to the changes in surface snow"**

In response to your comment, we have modified the following sentence:

Line 493: "Therefore, the change in surface snow is mainly attributed to changes in the surface precipitation."→ "Therefore, the change in surface precipitation is mainly attributed to changes in surface snow."

---

## Author Comment (AC2)

**Reviewer #2**

**General comments**

**1. This paper explored the effect of prognostic graupel density in WDM6. The goal of this modification was to better model snowfall events with more realistic fall speed-diameter and mass-diameter relationships. It was interesting to see assumptions in existing methods get replaced by theories in new studies with reasonable success (lower RMSE). However, the biggest concern I have with the comparison between WDM6_PD with WDM6_FD is that fall speed-diameter relationship does not converge when they have the same density (Figure 1). I understand that it might have been implemented this way because the authors tried to keep the off-the-shelf fixed density model unchanged, but this implementation fails to facilitate a fair comparison between fixed vs. prognostic density because no one knows if the difference in simulation results come from the prognostic density or the vastly different parameters (ag, bg). It also leads to physical inconsistency: starting L350 (also Figure 9), the graupel in WDM6_PD falls faster than WDM6_FD despite its lower prognostic density (250-350 kg/m3 vs 500 kg/m3 in WDM6_FD) and smaller size. I recommend adding a modified WDM6_FD that is simply WDM6_PD when rho= 500 kg/m3 " for a more meaningful comparison. This problem needs to be revisited because it could change the statistical skill scores shown in Table 4, but overall, this is a study worth publishing once this problem is resolved.**

: We appreciate your valuable comments. To explain the reason why the fall speed-diameter relationship between WDM6_PD and WDM6_FD does not converge when they have the same density, we added the following sentences in the revised manuscript:

Line 153: "Note that the coefficients, $a_G$ and $b_G$, are assumed as 330 $m^{1-b}\,s^{-1}$ and 0.8 in the original WDM6 scheme and these values differ significantly from those in Table 2. However, we adhere to the methodology presented in Milbrandt and Morrison (2013) to preserve the originality of the method."

Additionally, as the reviewer suggested, we conducted additional experiments to answer the question of whether the difference in simulation result comes from the prognostic density or the different parameters in the fall speed-diameter relationship. Specifically, we implemented $a_G$ and $b_G$ in the $V_G$–D relationship for $\rho_G = 500$ kg $m^{-3}$ in WDM6_PD from Table 1 (as shown by the blue line in Fig. R3) into the WDM6_FD scheme (referred to as WDM6_FD_500).

[Figure]

**Figure R3.** $V_G$ as a function of D for the $\rho_G$ range between 100 and 900 (kg $m^{-3}$). $a_G$ and $b_G$ in Table 1 are utilized. The $V_G$-D relationships in the WDM6_FD and WDM6_FD_500 are also represented by a red and blue lines.

The results, comparing WDM6_FD_500 with WDM6_FD for the CL case, are displayed in Figures R4 to R6. As mentioned in section 4.2 of the original manuscript, WDM6_PD leads to an increase in surface graupel, especially in regions where WDM6_FD exhibits a negative bias compared to AWS observations (Figs. 7a-d of the original manuscript). However, WDM6_FD_500 presents a different response of surface graupel compared. By presenting a reduction of surface graupel, WDM6_FD_500 increases the negative bias over the central to western part of analysis domain, despite the use of higher graupel fall velocity than in WDM6_FD (Fig. R4b). This leads to a deterioration of RSME score in WDM6_FD_500 (Table R1).

[Figure]

**Figure R4**. Accumulated surface precipitation amount (mm) for (a) CL case with WDM6_FD_500 during the analysis period. The differences in the amounts of surface precipitation (mm) between WDM6_FD_500 and WDM6_FD (WDM6_FD_500 minus WDM6_FD) for CL case are shaded in (b). The red (blue) solid lines represent the positive (negative) differences between WDM6_FD and AWS observations (WDM6_FD minus AWS). The contour lines for positive (negative) values are plotted at 3, 5, 7 and 10 (−3, −5, −7 and −10) mm. The differences in the amounts of surface snow (mm) between WDM6_FD_500 and WDM6_FD (WDM6_FD_500 minus WDM6_FD) for CL case are plotted in (c). The differences in the amounts of surface graupel (mm) are shown in (d).

**Table R1.** Statistical skill scores of the root mean square error (RMSE), bias, and equitable threat score (ETS) for the cases with WDM6_FD, WDM6_PD and WDM6_FD_500.

| Case | Experiment | RMSE | BIAS | ETS |
|------|-----------|------|------|-----|
| | WDM6_FD | 6.58 | 1.27 | 0.30 |
| CL | WDM6_PD | 6.01 | 1.61 | 0.31 |
| | WDM6_FD_500 | 6.50 | 1.19 | 0.31 |

WDM6_PD also exhibits a significant decrease in graupel mass mixing ratio in the atmosphere compared to WDM6_FD (Fig. 8c of the original manuscript), due to a relatively higher graupel fall velocity. However, WDM6_FD_500 (Fig. 5c), relative to WDM6_FD, shows a smaller reduction in graupel mass mixing ratio in the atmosphere, and the changes in other hydrometeor mixing ratios between WDM6_FD_500 and WDM6_FD are different from those between WDM6_PD and WDM6_FD, even though both WDM6_FD_500 and WDM6_PD have higher fall velocities than WDM6_FD. The analysis of the falling graupel mass mixing ratio (Fig. R6) reveals that the maximum level of falling graupel does not differ between WDM6_FD_500 and WDM6_FD. Meanwhile, Figure 9 of the original manuscript shows that the maximum level of falling graupel is simulated at a lower altitude in WDM6_PD compared to WDM6_FD.

[Figure]

**Figure R5.** Vertical profiles for the time-domain-averaged mass mixing ratios (g kg$^{-1}$) of hydrometeors for (a) CL case with WDM6_FD. (b) is same as (a), but for WDM6_WDM6_FD_500. The differences in the mass mixing ratios of WDM6_WDM6_FD_500 and WDM6_FD (WDM6_WDM6_FD_500 minus WDM6_FD) for CL case are plotted in (c). In (a) and (b), the cloud ice mass mixing ratio ($q_I$) is multiplied by 100. The sum of snow and graupel mass mixing ratios (g kg$^{-1}$) is indicated by red lines, and the 0°C level by the grey dashed horizontal line.

[Figure]

**Figure R6.** Time-domain averaged difference in graupel mass mixing ratio (g kg$^{-1}$) between the levels 'z' ($q_z$) and 'z-1' ($q_{z-1}$) due to sedimentation with WDM6_FD_500 and WDM6_FD for CL case is in (b). The solid and dashed lines represent WDM6_FD and WDM6_FD_500, respectively.

Consequently, the results of the additional experiment indicate that the findings presented in the manuscript for WDM6_PD are not solely attributable to higher fall velocity parameters when compared to WDM6_FD. Instead, the differences in simulation results originate from the predicted graupel density, which is calculated based on physically based microphysical processes, and the resulting variation in graupel fall velocity. The verification of the $\rho_G - V_G$ relationship using observed 2DVD data in Figure 11 of the original manuscript also suggests that WDM6_PD, with predicting graupel density and modified fall velocity, simulates a more realistic $\rho_G - V_G$ relationship compared to WDM6_FD.

**Specific comments**

2. **L138. Why are the parameters in Table 2 so far off from the ones in the original WDM6 scheme? I don't think you need an extensive explanation, but a one-sentence summary would be nice as it ties to points I made in the general comments.**

: In response to reviewer's suggestion, we have added the following sentences to the revised manuscript:

Line 153: "Note that the coefficients, $a_G$ and $b_G$, are assumed as $330 \text{ m}^{1-b}\text{s}^{-1}$ and 0.8 in the original WDM6 scheme and these values differ significantly from those in Table 2. However, we adhere to the methodology presented in Milbrandt and Morrison (2013) to preserve the originality of the method."

Additionally, to expand on this discussion and explore potential avenues for further refinement of the simulated fall velocities, we have added the following sentences in "Summary and Conclusion" section of the revised manuscript:

Line 505: "The derived $V_G$–D relationship in our research could be refined by incorporating a broader range of graupel observational data, including hexagonal, conical, lump graupel, or graupel-like snow. Improvements in the representation of $V_G$–D relationship can lead to better simulation of precipitation and microphysical processes in environments where various types of graupel are generated."

3. **L316. Briefly explain equitable threat score (ETS) and why it's worth mentioning (even though the scores are similar).**

: Thank you for your comment. Equitable Threat Score (ETS) is a valuable metric for assessing forecast performance because it accounts for random chance and provides a balanced measure of forecast skill. To explain why we mentioned the ETS score, we have added the following sentences to the revised manuscript:

Line 358: "Despite these similar ETS scores, this comparison confirms that both WDM6_FD and WDM6_PD perform comparably well in predicting snowfall events."

4. **L350. Physical inconsistency as mentioned in the general comments.**

: In response to comment #1, we have carefully reviewed your comments and conducted additional experiments to address the concerns you raised. We believe that additional experiments and followed description have effectively addressed the identified issues.

5. **L379. The enhanced graupel fall velocity should not be attributed to the prognostic graupel density but rather the vastly different parameters (ag, bg) used.**

: In response to your comment, we modified sentences as follows:

Line 427: "The significantly enhanced graupel fall velocity, attributed to the prognostic graupel density in WDM6_PD, accelerates the sedimentation of graupel." → "The significantly enhanced graupel fall velocity, attributed to the newly derived parameters in the $V_G$–D relationship in WDM6_PD, accelerates the sedimentation of graupel."

6. **L391. The slight enhancement of vertical velocity in the range of 0.1-0.5 m/s seems about equally insignificant for both CL and WL. The authors might also want to reexamine this figure with the WDM6_FD with modified parameters.**

: We concur with your assessment that the differences in vertical velocity depicted in Fig. 10 are minimal and do not substantiate the suggested impacts of microphysical changes as robustly as required for a conclusive argument. Considering your comments and after careful consideration, Fig. 10 and its related descriptions are removed in the revised manuscript.

---

## Author Comment (AC3)

**Reviewer #3**

This manuscript is focused on the development of a new prognostic parameter that controls graupel density and fall terminal speed in the WDM6 microphysics scheme. The 2-D idealized simulation of a squall line scenario was compared with observations from the ICE-POP 2018 field campaign during the Winter Olympics in Pyeongchang, Korea. Overall, the manuscript is well written and easy to follow. The reviewer has 3 main comments related to the impacts on cloud microphysical processes, the relationship with thermodynamic conditions specifically relative humidity, and the evaluation against more measurements. Reviewer recommends the manuscript being revised by addressing the following comments. Below are the main comments.

: Thank you for your constructive feedback. We have thoroughly addressed all comments raised by the reviewer in the revised manuscript. The detailed responses are noted as below.

1.   Figure 6 shows the evolution of graupel density and mixing ratio as well as the key source and sink terms. This figure is very helpful for understanding the formation of graupel. Can the authors also provide this figure for the WDM6_FD simulation? In addition, the reviewer suggests more discussion about the impacts and differences related to cloud microphysical processes. The authors can consider adding supplemental figures showing this type of cross sections in an evolutionary view for other variables, including cloud liquid, cloud ice, snow, and relative humidity with respect to ice.

: In response to the reviewer's comments, we have added Figure 7, presented graupel mass mixing ratio and key source and sink terms for the WDM6_FD, with the following additional sentences in the revised manuscript:

[Figure]

**Figure 7.** Same as Figure 6, but for WDM6_FD.

Line 317: "Same microphysical properties as in Figure 6, but for WDM6_FD, are shown in Fig. 7 except $\rho_G$. Note that $\rho_G$ in WDM6_FD is pre-defined as 500 kg m$^{-3}$. Throughout the simulation period, WDM6_FD produces a more abundant mass mixing ratio of graupel, reaching higher vertical levels and simulating a wider region for SUB (compare Figs. 7 and 6). At 2 h, graupel continues to be generated through DEP, ACC and FRZ, and the region with active SUB expands compared to the initial stage (Figs. 7c and d). At 4 h, more

graupel is transported into the anvil cloud region at relatively lower levels compared to WDM6_FD due to active DEP and ACC in the corresponding region (Figs. 7e and f)."

Additionally, as shown in the newly added flowchart of mass mixing ratio and number concentration (Fig. A1) in the appendix, the modified microphysics processes due to the predicted graupel density affect the evolution of other hydrometeors including graupel. We have included the following supplemental figures (Figs. S2-S5) depicting the cross-sections of cloud liquid, cloud ice, snow mass mixing ratio, and relative humidity with respect to ice (RHice) with the following sentences in the revised manuscript:

Line 324: "As we mentioned in section 2, varying parameters with the predicted graupel density can affect the mass mixing ratio and number concentration of other hydrometeors. The spatial distribution of the mass mixing ratio of other variables (cloud water, cloud ice, and snow) and the relative humidity with respect to ice (RHice) during the development stage of convection are available in Figures S2 to S5 of the supplement."

[Figure]

**Figure S2.** Spatial distribution of cloud water mass mixing ratio (g kg$^{-1}$) in WDM6_FD (a, c, and e) and WDM6_PD (b, d, and f) at 1 hour (a and b), 2 hour (c and d), and 4 hour (e and f).

[Figure]

**Figure S3.** Same as Figure 1, but for cloud ice.

[Figure]

**Figure S4.** Same as Figure 1, but for snow.

[Figure]

**Figure S5.** Same as Figure 1, but for relative humidity with respect to ice.

**2. Table 4 provides a critical assessment of the performance of the new scheme. However, there is not much description of this table. The review only found this sentence on line 310, stating that the RMSE for all CL cases is much improved. Is this a comparison of the graupel mass collected at surface?**

: We also explained the results of statistical scores for WL case as "The reduction in surface precipitation amount in WDM6_PD results in an improvement in the RMSE scores for all WL cases, as well as biases for all WL cases except for Case5 (Table 4)." in the original manuscript.

Additionally, to clarify what these statistical scores pertain to, we have revised the following sentence:

Line 371: "Statistical skill scores of the root mean square error (RMSE) (mm), bias (mm) and equitable threat score (ETS) for different cases with WDM6_FD and WDM6_PD." → "Statistical skill scores for surface precipitation, including the root mean square error (RMSE) (mm), bias (mm), and equitable threat score (ETS), for different cases with WDM6_FD and WDM6_PD."

**3. There are other measured properties that the authors mentioned in section 3.2, and line 220, which are the 2DVD measured diameter, fall velocity, and geometry of each hydrometeor falling. However, the reviewer didn't find comparisons on these three properties, or maybe they were mentioned only in the text but not in any figures and tables.**

: Thank you for your insightful comments. As mentioned in section 3.2 of the original manuscript, we used the 2DVD measured data of diameter, fall velocity, and geometry of hydrometeor to validate whether the model effectively reproduces the observation-derived density–fall velocity relationship of graupel. The 2DVD observed diameter and geometry of particles are used to identify graupel particles and estimate their density. Therefore, the 2DVD measured falling velocity ($V_G$) and derived graupel density ($\rho_G$) from diameter and geometry are used to verify the model simulated $\rho_G$–$V_G$ relationships, as shown in the Figure 12 of the revised manuscript.

**4. Can the authors provide additional comparison on these properties? In addition to surface measurements, are there other types of observations available? What about radiosonde measurements of temperature, humidity and wind speed, and satellite observations of cloud properties?**

: We appreciate your suggestion to provide additional comparisons of properties of the modified graupel species. We agree that more thoughtful verification with other observational data such as thermodynamic or hydrometeors profiles would be benefit the improvement of the model performance. However, our study put an effort on the development of new microphysics scheme containing predicting graupel density and its evaluation. Therefore, the verification of our study focusses on the $\rho_G$– $V_G$ relationships. As we added the discussion in the revised manuscript below, we will leave further verification with other observation for future works.

Line 505: "The derived $V_G$–D relationship in our research could be refined by incorporating a broader range of graupel observational data, including hexagonal, conical, lump graupel, or graupel-like snow. Improvements in the representation of $V_G$–D relationship can lead to better simulation of precipitation and microphysical processes in environments where various types of graupel are generated. Additionally, the potential benefits of the predicted graupel density could be further evaluated in future works through comparison with additional observational data such as sonde and satellite."

5. **Another main comment is about the relationship with thermodynamic conditions, especially with RHice. In previous studies evaluating other double moment microphysics schemes against in situ aircraft observations, the results showed large sensitivities of the cloud microphysical properties (I.e., ice crystal mass and number concentrations) to the ice nucleation thresholds in terms of RHice threshold. For example, the Morrison 2 moment scheme uses a default 108% RHice, and the Thompson 2 moment scheme uses a default 125% RHice for initiating ice nucleation at lower temperatures, which may lead to transition from clear sky ice supersaturation to ice crystals too early (D'Alessandro et al., 2017). And this RHice threshold was found to be more realistic if it is set at 125-130% in an idealized squall line scenario when compared with the NSF DC3 field campaign (Diao et al., 2017). The reviewer wonders what is the current ice nucleation threshold used in the WDM scheme, and if that parameter has a very large impact on the simulated graupel density and fall speed. If a similar RHice threshold is adopted in WDM scheme around 108% similar to the Morrison 2 moment scheme, then the reviewer recommends testing increasing that threshold to 125% or 130%.**

: Thank you for your insightful comments regarding the relationship with thermodynamic conditions, especially with RHice. In both WDM6_FD and WDM6PD, the RHice threshold is set at 108%. Based on the reviewer's suggestion, we have conducted additional 2D idealized squall line experiments with the RHice threshold increased to 130%, maintaining the same experimental setup as in the original manuscript. The results of this analysis are presented in Figure R7. The experiments are labeled as WDM6_PD with the original RHice value (PD108) and WDM6_PD with the increased RHice value (PD130).

Both simulations (PD108 and PD130) indicate that the simulated features of graupel density and its mass mixing ratio distributions are fairly consistent across the evolution time. The source and sink terms are also similar, though there are only minor differences such as a horizontal feature of sublimation (SUB) at 1 and 2 h between PD108 and PD130 (Figs. 6 and R7). These results implies that the choice of RHice threshold does not significantly affect the simulated graupel properties including its source/sink processes in the WDM6 microphysics scheme. Based on our simulation results and reviewer's comment, we have added the following sentence in the revised manuscript:

Line 328: "Meanwhile, Diao et al. (2017) suggested 125-130% of RHice threshold value is more realistic for an idealized squall line scenario when compared with the National Science Foundation (NSF) Deep Convective Clouds and Chemistry (DC3) field campaign. The increase of RHice from 108% to 130% in our 2D squall line setup does not affect the predicted graupel density features (not shown)."

[Figure]

**Figure R7**. Spatial distribution of $\rho_G$ (kg m$^{-3}$), $q_G$ (g kg$^{-1}$), and the major source/sink microphysics processes (g kg$^{-1}$) related to $q_G$ in PD130 at 1 hour (a and b), 2 hour (c and d), and 4 hour (e and f). In (b), (e), and (h), the solid red (blue) line represents positive (negative) vertical wind speed. Contour lines for positive (negative) value are at 2, 5, and 8 (−5 and −2). The unit of wind speed is m s$^{-1}$. In (c), (f), and (i), the main source processes are analyzed as Pgdep (DEP), the mean of Paacw, Psacr, and Pgacr (ACC), and Pgfrz (FRZ), while the sink ones are represented by sublimation (SUB) and Pgmlt (MLT). Red (blue) color shaded is represented as DEP (SUB). For the detail description of the microphysical processes are shown in Table1.

**Minor comment.**

**6. Some figures seem to have the dimension compressed, such as Figure 4, which seems to be compressed vertically (too narrow in vertical).**

**:** In response to the reviewer's comment, we have correctly adjusted the dimension of all figures including Figure 4 in the revised manuscript.

---

## Author Comment (AC4)

**Appendix: Description of WDM6 Microphysics Scheme**

**a) Parameters for hydrometeor characteristics**

[revised manuscript text omitted]

**Supplementary material:**

[Figure]

**Figure S1: Vertical profiles for the domain-averaged (a) sources and (b) sinks of graupel mass mixing ratio in WDM6_FD (Solid line) and WDM6_PD (dashed line) at 1 hour (a and b), 2 hour (c and d), and 4 hour (e and f). The main source processes, namely, deposition (Pgdep; DEP), accretion (mean of Paacw, Psacr and Pgacr; ACC) and freezing (Pgfrz; FRZ) are plotted with the major sink processes, namely, sublimation (Pgsub; SUB) and melting (Pgmlt; MLT).**

40

45

[Figure]

50    **Figure S2: Spatial distribution of cloud water mass mixing ratio (g kg⁻¹) in WDM6_FD (a, c, and e) and WDM6_PD (b, d, and f) at 1 hour (a and b), 2 hour (c and d), and 4 hour (e and f).**

[Figure]

**Figure S3: Same as Figure 1, but for cloud ice.**

[Figure]

55

**Figure S4: Same as Figure 1, but for snow.**

[Figure]

**Figure S5: Same as Figure 1, but for relative humidity with respect to ice.**

[Figure]

60

**Figure S6: Snow advection (g kg$^{-1}$ s$^{-1}$) and wind vector (m s$^{-1}$) at 850hPa for CL case from (a) WDM6_FD, and (b) the difference between WDM6_PD and WDM6_FD (WDM6_PD minus WDM6_FD). The vertical profiles of the time-domain-averaged Paacw process for C1 case are shown in (c). The solid (dashed) line represents WDM6_FD (WDM6_PD).**

65

---

## Author Response (AR2)

**[Response to the reviews]**

I commend the authors for their comprehensive revisions that have acted to greatly improve the manuscript and have highlighted the value of the sound science performed, presenting a foundation for further work to follow on from what was done here. Below are a few technical corrections and one more point (#2) that needs to be clarified.

: We appreciate the reviewer's valuable comments. We have thoroughly addressed all the comments raised by the reviewer in the revised manuscript. The detailed responses are noted below.

**1) When discussing the Best Number, need to use a different variable to represent it because "X" is already being used to represent hydrometeor species. Maybe a Greek Chi would work.**

: In response to the reviewer's comment, we have replaced "X" with "$\chi$" to represent the Best Number.

**2) The new quantitative statements on Lines 346-348 and 352-356 appear to be erroneous in some way. In lines 346-348, you state that surface graupel increases in WDM6_PD relative to WDM6_PD (by 124%), but then go on to stay that domain-averaged graupel amount is 0.64 mm in WDM6_FD and 0.51 mm in WDM6_PD, which would indicate a decrease in graupel for WDM6_PD relative to WDM6_FD.**

**Similarly, on lines 352-356, you state that surface snow decreases significantly (by 92%) in WDM6_PD, but then state that domain-averaged snow amount is 0.77 mm in WDM6_FD and 0.84 mm in WDM6_PD, which would indicate an increase; then for graupel, you say it increases by 121% in WDM6_PD, but then say that domain-averaged graupel is 0.21 mm in WDM6_FD and 0.18 mm in WDM6_PD, which would indicate a decrease. So something is wrong here unless I'm misunderstanding the interpretation.**

: Thank you for the thoughtful comment. In the previous version, we recorded the swapped values for graupel and snow amounts between the two experiments (WDM6_PD and WDM6_FD). We have modified the following sentences to correct the discrepancies.

-Line 346: "Specifically, the total surface snow is reduced by 93% (domain-averaged snow amount is 0.75 mm in WDM6_FD and 0.80 mm in WDM6_PD), and surface graupel shows an increase of 124% (domain-averaged graupel amount is 0.64 mm in WDM6_FD and 0.51 mm in WDM6_PD) in WDM6_PD compared to WDM6_FD."
→ "Specifically, the total surface snow is reduced by 93% (domain-averaged snow amount is 0.80 mm in WDM6_FD and 0.75 mm in WDM6_PD), and surface graupel shows an increase of 124% (domain-averaged graupel amount is 0.51 mm in WDM6_FD and 0.64 mm in WDM6_PD) in WDM6_PD compared to WDM6_FD."

-Line 352: "Surface snow decreases significantly by 92% in WDM6_PD (domain-averaged snow amount is 0.77 mm in WDM6_FD and 0.84 mm in WDM6_PD), compared to WDM6_FD, while the surface graupel increases by 121 % (domain-averaged graupel amount is 0.21 mm in WDM6_FD and 0.18 mm in. WDM6_PD) (Figs. 8g and h)." → "Surface snow decreases significantly by 92% in WDM6_PD (domain-averaged snow amount is 0.84 mm in WDM6_FD and 0.77 mm in WDM6_PD), compared to WDM6_FD, while the surface graupel increases by 121 % (domain-averaged graupel amount is 0.18 mm in WDM6_FD and 0.21 mm in. WDM6_PD) (Figs. 8g and h)."

**3) Line 451: I think "hights" should be "highlights"**
: Revised accordingly.

**4) Line 57: Picky semantics again, but P3 predicts the rime mass fraction and rime density, but rime volume is prognostic, not predicted.**
: In response to the reviewer's comment, we have modified the corresponding sentence as below:
-Line 57: "Morrison and Milbrandt (2015) later developed the Predicted Particle Properties (P3) bulk micro physics scheme that predicts the rime mass fraction,  and rime density for a single generic ice-phase category."